# ACCELERATED TRAINING THROUGH ITERATIVE GRADIENT PROPAGATION ALONG THE RESIDUAL PATH

**Erwan Fagnou**[1], **Paul Caillon**[1], **Blaise Delattre**[1,2] **& Alexandre Allauzen**[1,3]
[1] Miles Team, LAMSADE, Université Paris Dauphine-PSL, Paris, France
[2] Foxstream, Vaulx-en-Velin, France
[3] ESPCI PSL, Paris, France
`{name}.{surname}@dauphine.psl.eu`

## ABSTRACT

Despite being the cornerstone of deep learning, backpropagation is criticized for its inherent sequentiality, which can limit the scalability of very deep models. Such models faced convergence issues due to vanishing gradient, later resolved using residual connections. Variants of these are now widely used in modern architectures. However, the computational cost of backpropagation remains a major burden, accounting for most of the training time. Taking advantage of residual-like architectural designs, we introduce Highway backpropagation, a parallelizable iterative algorithm that approximates backpropagation, by alternatively i) accumulating the gradient estimates along the residual path, and ii) backpropagating them through every layer in parallel. This algorithm is naturally derived from a decomposition of the gradient as the sum of gradients flowing through all paths, and is adaptable to a diverse set of common architectures, ranging from ResNets and Transformers to recurrent neural networks. Through an extensive empirical study on a large selection of tasks and models, we evaluate Highway-BP and show that major speedups can be achieved with minimal performance degradation.

## 1 INTRODUCTION

Often copied but never matched, the backpropagation algorithm (Rumelhart et al., 1986) is still at the heart of deep-learning optimization, coupled with the gradient descent. However, as model sizes continue to grow, its memory overhead and computational time become more and more prohibitive. This was especially the case for recurrent neural networks (RNNs) (Elman, 1990; Hochreiter & Schmidhuber, 1997; Cho et al., 2014). Considered as state of the art for sequence processing (*e.g.* natural and spoken language), the time required to run the backpropagation through time for stacked RNNs (Sutskever et al., 2014) has motivated the design of transformers (Vaswani et al., 2017) that process the sequence in parallel. However, with the advent of deeper and larger models in NLP (Kaplan et al., 2020; Hoffmann et al., 2022) and computer vision (Dosovitskiy, 2020; Dehghani et al., 2023), the problem persists: the sequential aspect of backpropagation implies a computational cost that clearly limits further advancements in model design and scalability.

Frugal alternatives to backpropagation, such as forward-only methods (Hinton, 2022; Nøkland, 2016) and exact parallel backpropagation (Lim et al., 2024; Danieli et al., 2023), have shown promising results, but often involve impractical trade-offs between speed and task performance. Moreover, these methods often do not to leverage the recent advances that made the success of modern deep-learning models, like Batch and Layer-normalization (Ioffe & Szegedy, 2015; Ba et al., 2016). Another important example is the widespread use of residual connections, which enables efficient gradient propagation across layers, prevents vanishing gradients, and significantly improves training convergence in very deep models (Srivastava et al., 2015; He et al., 2016). Most contemporary deep models incorporate residual paths that connect the loss to intermediate layers.

In this work, we focus on deep sequential models, *i.e.* models that rely on a large and sequential computational graph like RNNs, ResNets, and Transformers. We introduce *Highway backpropagation (Highway-BP)*, an iterative algorithm to transmit the error signal backward through the network. Derived from an original approach, Highway-BP leverages residual paths to instantly backpropagate

gradient estimates to earlier layers. By varying the number of iterations, our method allows us to readily trade the level of approximation of the gradient for speed-up and precision, which lets the user choose a dedicated optimization strategy in the context of a limited computational budget.

Our main contributions are the following:

- We introduce Highway-BP, a parallelizable iterative algorithm that approximates backpropagation for accelerating the training of deep sequential models.
- By leveraging architecture-aware components such as residual connections, Highway-BP is highly efficient and can be directly adapted to many different architectures.
- The algorithm is motivated by an intuitive decomposition of the gradient. In particular, the speed versus accuracy trade-off of the algorithm can be controlled by stopping after $k$ iterations, resulting in an interpretable approximation.
- We evaluate Highway-BP on a large range of models and tasks, including ResNets, Transformers, and RNNs, and empirically show that it converges to the exact gradient in only a handful of iterations.

## 2 RELATED WORK

### 2.1 RESIDUAL CONNECTIONS

Gradient descent is the fundamental method for training deep learning models, but very deep networks encounter significant challenges, including vanishing and exploding gradients (Bengio et al., 1994; Pascanu et al., 2013; Zucchet & Orvieto, 2024). To address these problems, network architectures have been modified with connections that bypass intermediate layers. Residual connections, first introduced in ResNets (He et al., 2015) and later adopted in transformers (Vaswani et al., 2017; Radford & Narasimhan, 2018), are one such solution. Similar concepts are found in Highway networks (Srivastava et al., 2015) and the gated mechanisms of LSTMs (Hochreiter & Schmidhuber, 1997) and GRUs (Chung et al., 2014).

In addition to mitigating vanishing gradients, residual connections (Srivastava et al., 2015; He et al., 2016) help address the shattering gradient effect, where gradients in deep networks become noisy and uncorrelated, leading to poor signal-to-noise ratios during backpropagation (Balduzzi et al., 2017). By introducing shortcut paths that allow gradients to flow more effectively, residual connections preserve meaningful signals across layers and simplify learning by enabling networks to approximate identity mappings when needed. This facilitates the optimization of deep models and allows networks to scale in depth without suffering from performance degradation.

Veit et al. (2016) in particular observe that residual models actually behave like an ensemble of shallow models. They show that the gradient that goes through many residual connections has the most impact in the training. This is precisely the motivation behind our work, where we provide an algorithm to compute these gradients, faster than backpropagating through the entire model.

### 2.2 PARALLELIZING BACKPROPAGATION

**Exact parallel backpropagation** Backpropagation can be computed exactly in parallel, with complexity in $\mathcal{O}(\log_2 L)$, where $L$ is the number of layers. This is done using prefix scan algorithms (Hillis & Steele, 1986; Blelloch, 1990). However, while this seems attractive, there are serious limitations in practice since the algorithm involves i) computing the Jacobian matrices of all layers, and ii) many matrix-matrix multiplications, both of which are extremely time and memory-consuming. In particular, matrix-matrix multiplications lead to the algorithm's true time complexity being in $\mathcal{O}(Bd^3 \log_2 L)$ and memory in $\mathcal{O}(Bd^2 L)$, where $B$ is the batch size and $d$ the hidden dimension of the model. The cubic complexity with respect to the dimension completely prevents the use of this algorithm for large models. Still, DeepPCR and DEER (Danieli et al., 2023; Lim et al., 2024) obtained significant speedups for small-sized models. Gonzalez et al. (2024) also proposed ELK as a more stable and scalable improvement of DEER, in particular by approximating the Jacobians with diagonal matrices, which reduces the time and memory complexities to match that of backpropagation. Our method uses the same prefix scan algorithm but leverages the structure of the Jacobians to keep a low complexity.

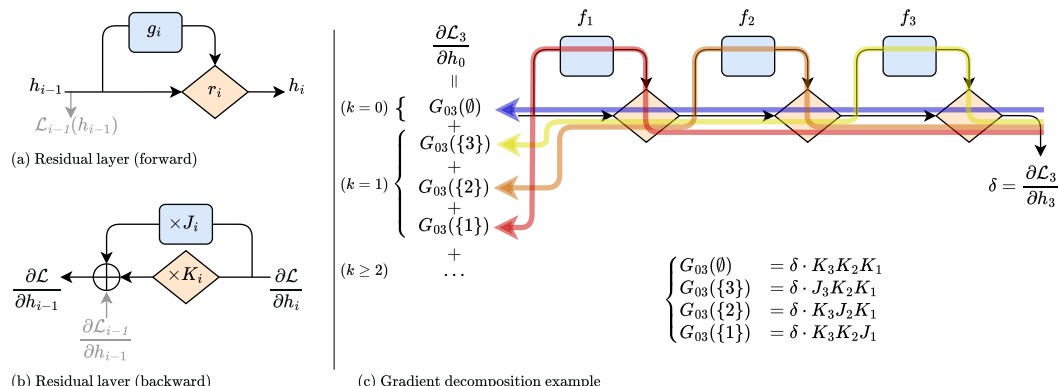

Figure 1: (a) Illustration of a layer $f_i$ decomposed as the composition of a block $g_i$ and a residual function $r_i$. (b) Backpropagation through the residual layer. Gray terms correspond to intermediate losses that are only present in RNNs. (c) Illustration of the gradient decomposition (Theorem 1) as the sum of the gradients flowing through only the residual connections ($k = 0$ in blue), through only one block ($k = 1$, in yellow, orange and red), and so on for $k = 2, 3, ...$ until $k = L$.

**Backpropagation as a system of equations**  Backpropagating through a sequential model can be seen as solving a system of equations of the form $\frac{\partial \mathcal{L}}{\partial h_i} = \frac{\partial \mathcal{L}}{\partial h_{i+1}} \frac{\partial h_{i+1}}{\partial h_i}, \forall i$. This leads to multiple works applying solvers to accelerate training. Günther et al. (2020) and Moon & Cyr (2022) use ODE interpretations of ResNets and GRUs respectively to use the parallel MGRIT solver (Falgout et al., 2014). Similarly, Wang & Ragni (2021) reformulate backpropagation in RNNs as finding a fixed point, which they compute iteratively. The method works well because each hidden state has its own local loss, which has more importance than losses further down the sequence. This however does not scale well to sequence classification and sequential models, which is why Trinh et al. (2018) introduce local auxiliary losses along the sequence, akin to pre-training in language modeling, to keep good performance even when truncating the gradients.

**Accelerating the forward pass**  Some of the aforementioned methods are also applied to approximate the forward pass: Lim et al. (2024); Danieli et al. (2023); Gonzalez et al. (2024); Wang & Ragni (2021); Günther et al. (2020). While this is out of the scope of this paper, our method is orthogonal as it only approximates backpropagation, and any method could be used concurrently to accelerate the forward pass.

Our approach differs from the system of equations and ODE interpretations, as we introduce Highway-BP through an intuitive decomposition of the gradient. Each element of the decomposition is progressively recovered at each iteration, giving a clear interpretation of the estimate after any $k$ iterations. Drawing inspiration from Danieli et al. (2023); Lim et al. (2024), we generalize their use of a scan algorithm and leverage architecture-aware components to make the computation much more efficient and scalable.

## 3 NOTATIONS AND ASSUMPTIONS

We consider a sequential model composed of $L$ layers $f_1, f_2, \ldots, f_L$, each parameterized by $\theta_i$. We denote $h_i = f_i(h_{i-1})$ as the hidden state after layer $i$, with $x = h_0$ being the input of the model. We also define a loss function $\mathcal{L}$ to be minimized.

While most of the time the loss is only a function of the last hidden state $h_L$, we build our framework using a more general formulation, with a loss of the form: $\mathcal{L}(h_1, \ldots, h_L) = \sum_{i=1}^{L} \mathcal{L}_i(h_i)$. Allowing the loss to depend on each intermediate state allows the framework to handle more models and tasks (*e.g.* RNNs and transformers).

**Main assumption.** *We suppose that the layers $f_i$ can be expressed as the composition of two functions $g_i$ and $r_i$:*

$$f_i(x) = r_i(x, g_i(x)) \tag{1}$$

*leading to the following Jacobian:*

$$\frac{\partial f_i}{\partial x} = J_i + K_i \tag{2}$$

*where:*

- $J_i = \frac{\partial r_i(x,z)}{\partial z}\frac{\partial g_i(x)}{\partial x}$ *is **computationally expensive** to compute and can only be multiplied by a vector (e.g. Jacobian of a convolution),*

- $K_i = \frac{\partial r_i(x,z)}{\partial x}$ *is **computationally cheap** to compute and multiply (e.g. a diagonal matrix).*

This general formulation allows us to consider a wide range of architectures and models. For instance, in a simple residual model, we would choose $g_i$ to be the residual block (*e.g.* convolutions), and $r_i(x,z) = x + z$ would be the residual connection. Table 1 provides more examples including ResNets, Transformers (either pre- or post-layernorm), as well as recurrent neural networks like GRU and LSTM. Figure 1 also illustrates the decomposition of $f_i$ into $g_i$ and $r_i$.

Note that there is no requirement for the residual function $r_i$ to be linear. For instance, ResNets use a ReLU activation after the residual connection. This opens up our method to a wider class of models that may have more elaborate residual-like connections. Throughout the paper, we will designate any such architectural design as a residual connection or residual path.

Table 1: Examples of decomposition of the layers $f_i$ of common models as the composition of an expensive block $g_i(x)$ and a cheap residual connection $r_i(x,z)$, as in Equation 1.

| Model | $f_i(x)$ | $g_i(x)$ | $r_i(x,z)$ |
|---|---|---|---|
| Pre-activation ResNet | $x + \text{Block}(x)$ | $\text{Block}(x)$ | $x + z$ |
| ResNet | $\text{ReLU}(x + \text{Block}(x))$ | $\text{Block}(x)$ | $\text{ReLU}(x + z)$ |
| Transformer (Pre-LN) | $x + \text{Layer}(\text{LN}(x))$ | $\text{Layer}(\text{LN}(x))$ | $x + z$ |
| Transformer (Post-LN) | $\text{LN}(x + \text{Layer}(x))$ | $[a(x), b(x)]$ | $z_1 \odot x + z_2$ |
| GRU | $a(x) \odot x + b(x)$ | $[a(x), b(x)]$ | $z_1 \odot x + z_2$ |
| LSTM | $[a(x) \odot x + b(x), c(x)]$ | $[a(x), b(x), c(x)]$ | $[z_1 \odot x + z_2, z_3]$ |

## 4 HIGHWAY BACKPROPAGATION

Our method is based on a gradient decomposition into terms corresponding to different paths (Theorem 1). Based on a recursive relation, we introduce an iterative algorithm, which progressively includes gradients from longer paths (Theorem 2). Finally, we describe how an iteration can be parallelized. We provide proofs of the theorems in Appendix A.

### 4.1 GRADIENT DECOMPOSITION

At each layer $f_i$ one part of the gradient is backpropagated through the residual block (using $J_i$) and the other through the residual connection (using $K_i$). This leads to $2^L$ possible paths. The following theorem states that the gradient $\frac{\partial \mathcal{L}}{\partial h_i}$ is the sum of the gradients backpropagated through each path. The different paths are depicted in Figure 1.

**Theorem 1** (Decomposition of the gradient over all paths)**.** *Given two layer indices $i \leq j$, and a set of indices $\mathcal{J} \subseteq [i+1, j]$, we define $G_{ij}(\mathcal{J})$ as the gradient backpropagated from $\mathcal{L}_j(h_j)$ to $h_i$, going through either the Jacobian $J_k$ of the residual blocks (when $k \in \mathcal{J}$) and otherwise through the residual connections with $K_k$ (see Figure 1 for a visual example). It can be expressed as:*

$$G_{ij}(\mathcal{J}) := \frac{\partial \mathcal{L}_j}{\partial h_j} \prod_{k=0}^{j-i-1} \left( \begin{cases} J_{j-k} & \text{if } j-k \in \mathcal{J} \\ K_{j-k} & \text{otherwise} \end{cases} \right). \tag{3}$$

*Then, for any hidden state $h_i$, its gradient $\frac{\partial \mathcal{L}}{\partial h_i}$ is the sum over all paths starting at index $i$:*

$$\frac{\partial \mathcal{L}}{\partial h_i} = \sum_{\substack{i \leq j \leq L \\ \mathcal{J} \subseteq [i+1,j]}} G_{ij}(\mathcal{J}). \tag{4}$$

## 4.2 ITERATIVE ALGORITHM

Using the previous decomposition of the gradient, we can design an iterative process to compute it, as described in the following theorem:

**Theorem 2** (Iterative computation of the gradient). *Let us define $w_i^k$ as the sum of the gradients of all paths starting from $i$ going through at most $k$ Jacobians $J_j$, which we obtain by truncating the sum in Equation 4:*

$$w_i^k := \sum_{\substack{i \leq j \leq L \\ \mathcal{J} \subseteq [i+1,j] \\ |\mathcal{J}| \leq k}} G_{ij}(\mathcal{J}) \tag{5}$$

*Then, $w_i^k$ can be computed iteratively using the recursive relation:*

$$\begin{cases} w_i^0 &= \sum\limits_{j=i}^{L} \frac{\partial \mathcal{L}_j}{\partial h_j} K_j K_{j-1} \dots K_{i+1} \tag{6} \\[2em] w_i^{k+1} &= w_i^0 + \sum\limits_{j=i+1}^{L} w_j^k J_j K_{j-1} K_{j-2} \dots K_{i+1} \tag{7} \end{cases}$$

*In particular, for $k \geq L - i$ we get the exact gradient $w_i^k = \frac{\partial \mathcal{L}}{\partial h_i}$.*

Following this theorem, the algorithm requires at most $k = L$ iterations. It is then straightforward to finalize backpropagation and get the gradient with respect to the parameters: $\frac{\partial \mathcal{L}}{\partial \theta_i} = \frac{\partial \mathcal{L}}{\partial h_i} \frac{\partial f_i}{\partial \theta_i}$.

## 4.3 PARALLEL COMPUTATION

Breaking down Equation 7, we can see how an iteration can be computed in two steps:

1. A parallel backpropagation through the expensive Jacobians $J_i$:

$$v_i^{k+1} = w_{i+1}^k J_{i+1} \qquad \forall i \in [0, L-1] \tag{8}$$

2. A sequential backpropagation through the residual path, which is also parallelizable efficiently given our assumptions about $K_i$: $w_i^{k+1} = w_i^0 + u_i^{k+1}$, where:

$$u_i^{k+1} = \sum_{j=i}^{L-1} v_j^{k+1} K_j K_{j-1} \dots K_{i+1} = v_i^{k+1} + u_{i+1}^{k+1} K_{i+1} \tag{9}$$

While it is clear that step 1 is parallelizable, this is less obvious in step 2. This is however possible using prefix scan algorithms (Blelloch, 1990; Boehm et al., 2019). A prefix scan aggregates a series of values (*e.g.* vectors) using an associative operator (*e.g.* sum), which is a general formulation that has many applications, including solving linear recurrences like the one we have in Equation 9. We use Hillis and Steele's parallel algorithm (Hillis & Steele, 1986) in our experiments, and we indicate a pseudocode of this algorithm adapted to our needs in Appendix B (Algorithm 1). We denote this algorithm as CumSumProd as in Boehm et al. (2019), and use it to rewrite Equations 6 and 7:

$$\begin{cases} w_i^0 &= \text{CumSumProd}\left( \left( \frac{\partial \mathcal{L}_i}{\partial h_i} \right)_{i=1}^{L}, (K_i)_{i=1}^{L} \right)_i \tag{10} \\[2em] w_i^{k+1} &= w_i^0 + \text{CumSumProd}\left( \left( w_{i+1}^k J_{i+1} \right)_{i=1}^{L-1}, (K_i)_{i=1}^{L-1} \right)_i \tag{11} \end{cases}$$

$$\text{with:} \quad \text{CumSumProd}(a, M)_i := \sum_{j \geq i} a_j M_j M_{j-1} \dots M_{i+1} \tag{12}$$

On a single process, the parallel version of CumSumProd has a $\mathcal{O}(L \log L)$ time complexity, however as the inner loop is parallelizable the effective computation time grows in $\mathcal{O}(\log L)$. In addition, it can be implemented using in-place operations for optimal memory efficiency.

An important note is that the parallel CumSumProd algorithm relies on all the $K_i$ being computed ahead, and involves matrix-matrix multiplications between the $K_i$. This is not an issue in our case

Table 2: Summary of the models and datasets used in our experiments. $L$ is the number of skip-connections for sequential models, and the sequence length for RNNs. Only some RNNs have intermediate losses (*i.e.* one loss per cell)

| Dataset | Model | Layers | Params | $L$ | Intermediate losses |
|---|---|---|---|---|---|
| CIFAR10 | Pre-act ResNet32 | 32 | 464k | 15 | ✗ |
| CIFAR10 | ResNet110 | 110 | 1.7M | 54 | ✗ |
| ImageNet32 | ResNet56 | 56 | 917k | 27 | ✗ |
| Wikitext103 | GPT-2 | 12 | 14.5M | 24 | ✗ |
| MNLI | RoBERTa | 12 | 124M | 24 | ✗ |
| Wikitext103 – char | LSTM | 1 | 2.3M | 256 | ✓ |
| Wikitext103 – char | GRU | 1 | 1.8M | 256 | ✓ |
| Wikitext103 | GRU | 3 | 21.1M | 256 | ✓ |
| CIFAR10 – pixel level | GRU | 1 | 29.8k | 1024 | ✗ |

as we decompose the layer $f_i$ into $g_i$ and $r_i$ precisely such that $K_i$ behaves nicely (*e.g.* scalar, identity, diagonal, low-rank). Lim et al. (2024) also use CumSumProd but replace $K_i$ with the Jacobian of the whole layer, which is extremely inefficient in time and memory for large models. Also note that Highway-BP could still be applied to situations where $K_i$ prevents the use of the parallel CumSumProd algorithm, as it is always possible to solve sequentially the recursive relation in Equation 9, which only involves vector-Jacobian products.

## 4.4 APPROXIMATING BACKPROPAGATION

While the iterative process from Theorem 2 converges to the exact gradient after $L$ iterations, we propose to stop after a small number $k$ of iterations and use the current estimate $w^k$ instead of the exact gradient $w^L$ to update the model's parameters.

The number $k$ of Highway-BP iterations becomes a hyperparameter, and allows users to freely control the tradeoff between the speed and accuracy of the algorithm. Moreover, at any iteration $k$ the current estimate is interpretable by design: $w^k$ is the sum of all gradients flowing through at most $k$ residual blocks.

It is reasonable to expect that gradients going through fewer blocks are statistically more useful for learning. The reason why residual connections are so effective at improving training is that gradients can flow directly from the loss to any intermediate layer. All layers can learn at the same time, which greatly improves convergence. This suggests that the most important part of the gradient comes from the residual connection (or at least at the beginning of the training). Veit et al. (2016) have shown that this is the case for ResNet models. We also empirically confirm this throughout all of our experiments, described in section 5.

This leads to Highway-BP only requiring $k$ iterations, each having $\mathcal{O}(\log_2 L)$ substeps for the CumSumProd operation, thus reducing the computation time $\mathcal{T}$ from $\mathcal{T}_{\text{BP}} = \mathcal{T}_{\text{forward}} + \mathcal{T}_{\text{backward}}$ to:

$$\mathcal{T}_{\text{Highway-BP}} = \mathcal{T}_{\text{forward}} + \frac{k}{L}\mathcal{T}_{\text{backward}} + \mathcal{O}(k \log_2 L) \tag{13}$$

## 5 EXPERIMENTS

The Highway-BP framework and notations have been designed to be highly flexible, and in particular to handle both deep sequential models and recurrent neural networks. We evaluate Highway-BP on such models for several tasks, which we summarize in Table 2.

### 5.1 DEEP SEQUENTIAL MODELS

We evaluate Highway-BP on image classification with three ResNet models, as well as language modeling tasks by pre-training and fine-tuning two transformer models. The models greatly vary in size and depth, ranging from 464k to 124M parameters.

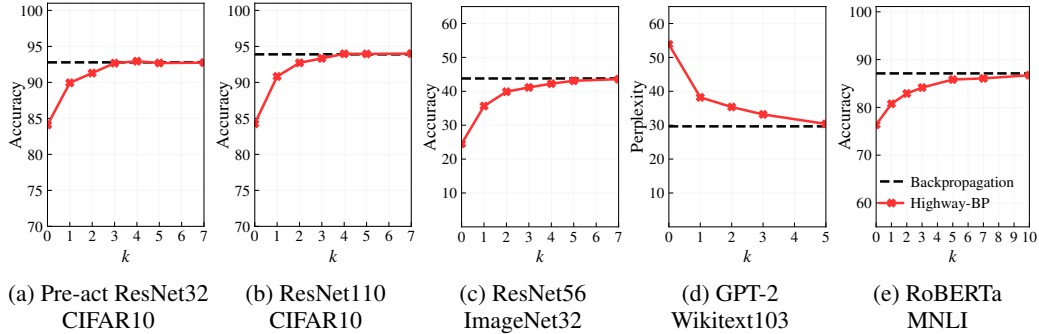

(a) Pre-act ResNet32 CIFAR10     (b) ResNet110 CIFAR10     (c) ResNet56 ImageNet32     (d) GPT-2 Wikitext103     (e) RoBERTa MNLI

Figure 2: Final performance of deep sequential models versus the number $k$ of Highway-BP iterations used for training (red), compared to backpropagation (black).

### 5.1.1 EXPERIMENTAL SETUP

The first ResNet model is a ResNet32 with pre-activations as introduced by He et al. (2016). It is easy to handle since we have $r_i(x, z) = x + z$ and $K_i = I$. We also use deeper ResNets with the original architecture (He et al., 2015), which apply a ReLU activation after the residual connection: $r_i(x, z) = \text{ReLU}(x + z)$. We train these models on CIFAR10 (Krizhevsky, 2009) and ImageNet32 (Chrabaszcz et al., 2017) for image classification. We only modified the downsampling layers as described in Appendix G to simplify the use of Highway-BP.

We also pre-train a small transformer model (Vaswani et al., 2017) for language modeling on the Wikitext103 dataset. The model is based on the GPT-2 architecture (Radford et al., 2019) with 12 layers but a smaller hidden dimension. Finally, we fine-tune a pre-trained RoBERTa model (Liu et al., 2019) on the MNLI dataset (Williams et al., 2018), which involves predicting the entailment information of a pair of sentences and is part of the GLUE benchmark (Wang et al., 2018). Note that for both transformers, we split the layers into two sublayers – self-attention and feedforward – which means we have $L = 24$ for 12 layers.

When applying Highway-BP to any of these models, we define $g_i$ and $r_i$ as described in Table 1. However, for both transformer models, we used slightly different choices as described in Appendix C.3. This is done after observing that transformer layers tend to learn to cancel part of their residual connection.

### 5.1.2 RESULTS

We report the results in Figure 2, where we compare models trained either with backpropagation or with Highway-BP using different numbers of iterations. As expected, more iterations increase the performance of the models. However, the quality of training is good even with very small values of $k$ compared to $L$. This is especially impressive for the ResNet110 model, which requires only $k = 4$ iterations to match backpropagation, while $L = 54$. This confirms our intuition that most of the gradient in deep residual models goes through the residual layers.

Surprisingly, performing $k = 0$ iterations already leads to very reasonable performances (*e.g.* 85% on CIFAR10). By definition of Highway-BP using Equation 4, the estimate after $k = 0$ corresponds to only backpropagating the gradient from the classification head through the residual path. Each layer then receives the gradient $\frac{\partial \mathcal{L}}{\partial h_L}$ at its output, and uses this to update its weights. This is very similar to boosting (Freund & Schapire, 1999), where many small models are summed together, and each one of them learns to compensate for the errors of the previous models.

The training curves of the GPT-2 model are shown in Figure 3, for different values of $k$. Even when $k$ is too low and deteriorates the model's performance, it still makes the model learn smoothly at the same speed, only converging to a higher loss.

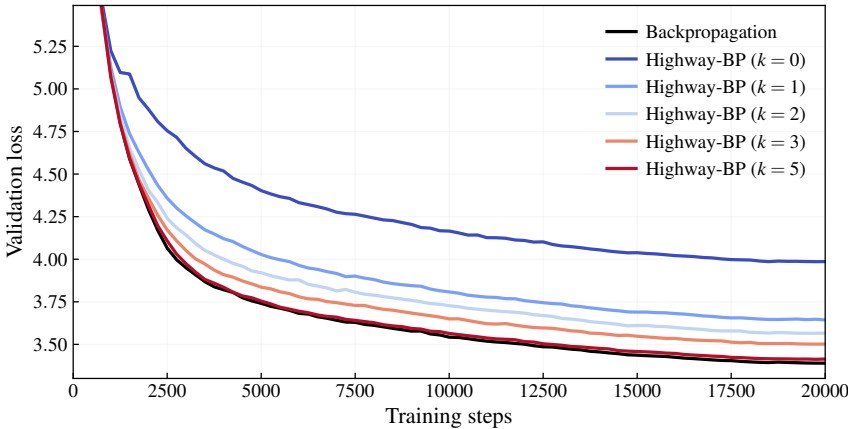

Figure 3: Validation loss during training for the GPT-2 model, with different algorithms.

## 5.2 RECURRENT NEURAL NETWORKS

Recurrent neural networks (RNNs) also fit our framework: instead of considering a sequence of layers along the depth dimension, we consider a repetition of the same cell along the time dimension. More formally, for an input sequence $(x_i)_{i=1}^L$, we can see each cell as a layer $f_i$ parameterized by $\theta_i = [\theta, x_i]$ (the parameter $\theta$ common to all cells, and the external input $x_i$). The input state $h_0$ is the initial state of the RNN.

Some RNNs such as LSTM (Hochreiter & Schmidhuber, 1997) and GRU (Chung et al., 2014) possess a long-term memory that is updated by each cell using a linear gating. This memory allows the model to keep information for long distances and helps with gradient issues (Zucchet & Orvieto, 2024). Akin to residual connections in deep models, we can take advantage of this architecture design with Highway-BP. We show in Table 1 how LSTM and GRU cells can be represented using the $g_i$ and $r_i$ functions.

### 5.2.1 EXPERIMENTAL SETUP

As baselines to compare the performance of Highway-BP on RNNs, we use i) backpropagation, and ii) fixed-point iteration (FPI), which is the method used by Wang & Ragni (2021) to approximate the backward pass of RNNs, and simply consists of repeating $k$ backpropagations through all layers in parallel. FPI is a special case of Highway-BP with $g_i = f_i$ and $r_i(x, z) = z$. Note that this algorithm can only perform well if there are intermediate losses at each time step, otherwise, this is equivalent to backpropagating only through the last $k$ cells, which can be seen as a form of extreme machine learning (Huang et al., 2006).

We train one layer of LSTM and GRU on a language modeling task at the character level on Wikitext103, as well as 3 GRU layers stacked trained on Wikitext103 at the word level (same task as the GPT-2 transformer in the previous section). Finally, we use a task from Long Range Arena (Tay et al., 2021): image classification on CIFAR10 using the flattened image, *i.e.* a sequence of 1024 3-dimensional pixel vectors.

### 5.2.2 RESULTS

Similarly to sequential models, we show in Figure 4 the performances of models trained with different algorithms, and for different numbers of iterations $k$. Highway-BP constantly outperforms the fixed-point iteration algorithm in terms of convergence speed, while the algorithms are practically identical in terms of computations performed. As mentioned in the previous section, FPI is a special case of our method when we do not consider the residual connection at all ($g_i(x) = f_i(x)$ and $r_i(x, z) = z$). Highway-BP uses additional knowledge about the architecture to improve the convergence speed over naive, model-agnostic approaches.

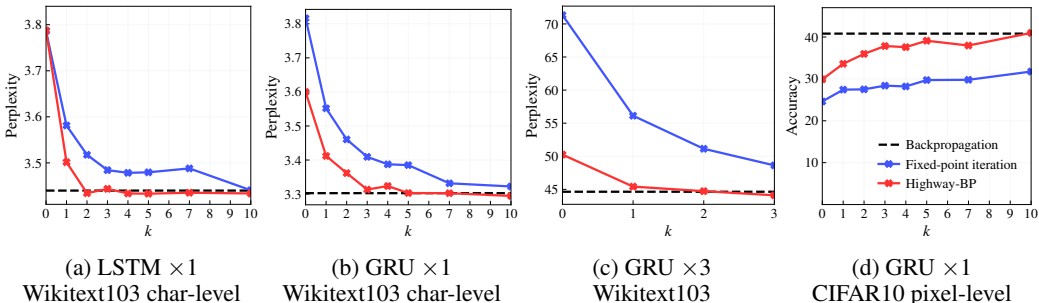

(a) LSTM $\times 1$
Wikitext103 char-level

(b) GRU $\times 1$
Wikitext103 char-level

(c) GRU $\times 3$
Wikitext103

(d) GRU $\times 1$
CIFAR10 pixel-level

Figure 4: Final performance of RNNs versus the number $k$ of Highway-BP iterations used for training (red), compared to backpropagation (black) and fixed-point iteration (blue).

Table 3: Speedup of training with Highway-BP vs. backpropagation for the RNN experiments (more details in Table 6).

| Model | $L$ | $k = 0$ | $k = 5$ | $k = 10$ |
|---|---|---|---|---|
| **LSTM** $\times 1$ | 256 | $\times 3.0$ | $\times 1.7$ | $\times 1.2$ |
| **GRU** $\times 1$ | 256 | $\times 3.2$ | $\times 1.8$ | $\times 1.3$ |
| **GRU** $\times 3$ | 256 | $\times 2.9$ | $\times 1.8$ | $\times 1.3$ |
| **GRU** $\times 1$ | 1024 | $\times 3.5$ | $\times 3.1$ | $\times 2.9$ |

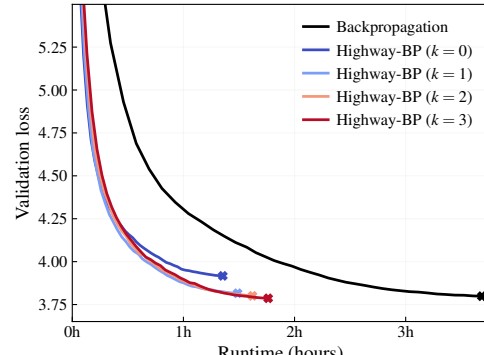

Figure 5: Loss vs. runtime of the 3-layer GRU RNN, for different algorithms.

The pixel-level CIFAR10 is an especially hard task, as the model needs to learn features from all parts of a very long sequence (1024 pixels). Moreover, the gradient is sparse as the prediction is made using the last hidden state $h_{1024}$, which means the gradient needs to be backpropagated through long distances. While FPI fails to do so as expected, Highway-BP reaches the same accuracy as backpropagation with $k = 10$ iterations.

We additionally report the speedup obtained with Highway-BP in Table 3. We observe that using the optimal number of iterations, all model trainings get a speedup between $\times 2$ and $\times 3$. Moreover, the gains get more significant for longer sequences.

Finally, in Figure 5 we show the training curves of the largest RNN model, the 3-layer GRU, using the real training time for the x-axis. It can be seen how $k$ controls the tradeoff between training speed and model performance.

## 5.3 TRAINING DYNAMICS ANALYSIS

In this section, we investigate how the convergence of Highway-BP evolves during training. Intuitively, at initialization, all the layers start learning using the residual connection. As the layers start using relevant features from earlier layers, they start working together and we expect the contribution of high values of $k$ to increase over training.

In Figure 6, we analyze how Highway-BP behaves throughout training. The top row reports the cosine similarity between the estimated gradient and the true gradient, which seems to require more iterations at the end of training to reach 1. The bottom row also validates this claim, as it shows how the contribution of each iteration slowly shifts toward higher values of $k$. The transformer seems to be the most consistent model, as the cosine similarity stays mostly constant during training.

The special case $k = 0$, which corresponds to only backpropagating through the residual connection, slowly decreases in accuracy over time for all models. Still, its contribution to the total norm remains

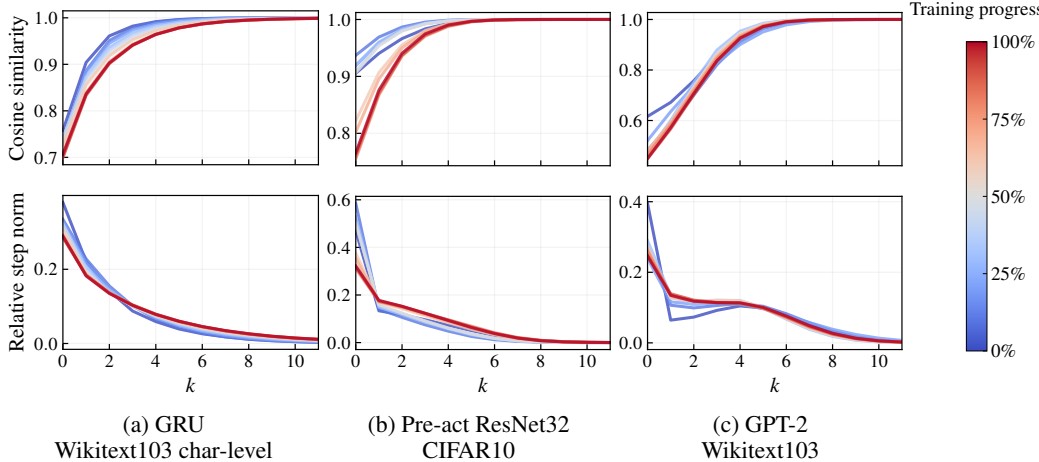

Figure 6: Convergence analysis of Highway-BP for different number $k$ of iterations and at different times during training (from 0% in blue, to 100% in red). (Top) Cosine similarity between the true gradient $\frac{\partial \mathcal{L}}{\partial \theta}$ versus the approximation using $k$ Highway-BP iterations. (Bottom) Norm of the $k$-th Highway-BP iteration step (relative to the total norm).

high during all the training, which suggests that residual connections play a key part in training deep models, and not only at the beginning of the training.

## 6  DISCUSSION AND CONCLUSION

We introduce Highway-BP as an architecture-aware algorithm for approximating backpropagation in deep sequential models. Its parallelizability and effectiveness unlock new possibilities, for instance, in training RNNs over long sequences or large models in a layer-distributed setting. We show through a decomposition of the gradient that each algorithm iteration adds another component to the estimate, until it completely reconstructs the gradient. As such, each intermediate estimate is interpretable as the sum of the gradients associated with paths going through at most $k$ residual blocks.

Empirical findings show that our method can replicate backpropagation with much lower time complexity, as it often converges in a few iterations. We observe this for all models, with promising results on deep models and RNNs on long sequences. As recently shown by Beck et al. (2024), when scaling LSTMs to billions of parameters, these models perform favorably in terms of performance compared to state-of-the-art Transformers, showcasing superior expressivity. Our framework may thus be applied to allow fast training of very large RNNs, with billions of parameters.

Our general framework allows us to use the same generic code to train all models using Highway-BP. However, simplicity comes at the cost of less optimization, and we believe that architecture-specific implementations must be done to benefit the most from Highway-BP. In addition, our main purpose in this paper is to demonstrate the high training quality of Highway-BP, almost matching backpropagation with a few iterations. We leave its practical implementation for training large models in a distributed setting for future work. We however show that RNNs can be sped up considerably on a single GPU, as all cells share the same weights. Still, we believe the prefix scan algorithm could be much more optimized, using a custom CUDA kernel for instance.

Finally, the tradeoff between training speed and quality can be adjusted at any time. While low numbers of iterations are enough at the beginning of training, it is possible to increase the number of iterations during training and end with an exact backpropagation. This versatility not only allows us to perform increasingly more accurate optimization steps to speed up training while attaining the same performance, but also allows the user to choose a dedicated optimization strategy in the context of a limited computational budget.

ACKNOWLEDGMENTS

This work was performed using HPC resources from GENCI-IDRIS (grants AD011015154 and A0151014627), and received funding from the French National Research Agency (ANR SPEED-20-CE23-0025) and the French Government via the program France 2030 ANR-23-PEIA-0008, SHARP.

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

## A  PROOFS

### A.1  PROOF OF THEOREM 1

The result is obtained by completely expanding the product of Jacobians obtained with the chain rule:

$$
\frac{\partial \mathcal{L}}{\partial h_i} = \sum_{j=i}^{L} \frac{\partial \mathcal{L}_j}{\partial h_j} \frac{\partial h_j}{\partial h_i}
$$

$$
= \sum_{j=i}^{L} \frac{\partial \mathcal{L}_j}{\partial h_j} (J_j + K_j)(J_{j-1} + K_{j-1}) \dots (J_{i+1} + K_{i+1})
$$

$$
= \sum_{j=i}^{L} \sum_{\mathcal{J} \subseteq [i+1,j]} \frac{\partial \mathcal{L}_j}{\partial h_j} \prod_{k=0}^{j-i-1} (J_{j-k} \text{ if } j-k \in \mathcal{J} \text{ else } K_{j-k})
$$

$$
= \sum_{\substack{i \leq j \leq L \\ \mathcal{J} \subseteq [i+1,j]}} G_{ij}(\mathcal{J})
$$

### A.2  PROOF OF THEOREM 2

From the definition of $w_i^k$:

$$
w_i^{k+1} = \sum_{\substack{i \leq j \leq L \\ \mathcal{J} \subseteq [i+1,j] \\ |\mathcal{J}| \leq k+1}} G_{ij}(\mathcal{J})
$$

$$
= \sum_{j=i}^{L} G_{\emptyset}^{ij} + \sum_{\substack{i \leq j \leq L \\ \mathcal{J} \subseteq [i+1,j] \\ 1 \leq |\mathcal{J}| \leq k+1}} G_{i\,\min(\mathcal{J})}(\mathcal{J} \setminus \{\min(\mathcal{J})\}) J_{\min(\mathcal{J})} K_{\min(\mathcal{J})-1} \dots K_{i+2} K_{i+1}
$$

$$= w_i^0 + \sum_{m=i+1}^{L} \left( \sum_{\substack{m \leq j \leq L \\ \mathcal{J} \subseteq [m+1,j] \\ |\mathcal{J}| \leq k}} G_{mj}(\mathcal{J}) \right) J_m K_{m-1} \dots K_{i+2} K_{i+1}$$

$$= w_i^0 + \sum_{j=i+1}^{L} w_j^k J_j K_{j-1} \dots K_{i+2} K_{i+1}$$

In particular, from Theorem 1 we have that $w_i^k = \frac{\partial \mathcal{L}}{\partial h_i}$ for $k \geq L - i$.

# B  PSEUDOCODE OF THE PARALLEL PREFIX SCAN ALGORITHM FOR CUMSUMPROD

---

**Algorithm 1 (Parallel CumSumProd)** Parallel cumulative sum-product algorithm (reversed)

---

**Inputs:**
  a sequence of $L$ vectors $(v_i)_{i=1}^{L}$ with $v_i \in \mathbb{R}^{d_i}$,
  a sequence of $L$ matrices $(K_i)_{i=1}^{L}$ with $K_i \in \mathbb{R}^{d_i \times d_{i-1}}$
**Output:** $(u_i)_{i=1}^{L}$ such that $u_i = \sum_{j=i}^{L} v_j K_j \dots K_{i+1}$

  $u^{(0)} \leftarrow v$
  $P^{(0)} \leftarrow K$
  $M \leftarrow \lceil \log_2 L \rceil$
  **for** $m \leftarrow 0$ **to** $M - 1$ **do**
    **for** $i \leftarrow 1$ **to** $L$ **in parallel do**
      **if** $i \leq L - 2^m$ **then**
        $u_i^{(m+1)} \leftarrow u_i^{(m)} + u_{i+2^m}^{(m)} \cdot P_i^{(m)}$
        $P_i^{(m+1)} \leftarrow P_{i+2^m}^{(m)} \cdot P_i^{(m)}$
      **else**
        $u_i^{(m+1)} \leftarrow u_i^{(m)}$
        $P_i^{(m+1)} \leftarrow P_i^{(m)}$
      **end if**
    **end for**
  **end for**
  **return** $u^{(M)}$

---

# C  EXPERIMENTAL SETUP DETAILS

## C.1  TRAINING SCHEME

Most models are trained using the Adam optimizer (Kingma & Ba, 2014). In case of weight decay, we use the AdamW variation (Loshchilov & Hutter, 2017). We also use a cosine learning rate scheduler to decrease the learning rate to a tenth of its initial value. Additionally, the first 10% of the training is performed with a linear warmup. For both ResNet experiments, however, we use a training scheme close to the original paper (He et al., 2015), which uses an SGD optimizer with momentum and divides the learning rate by 10 twice during training. All experiments were conducted on single GPUs, either Nvidia A100, A40, or RTX A6000.

## C.2  DATASETS

**CIFAR10**    The CIFAR10 (Krizhevsky, 2009) dataset contains 50k images with 10 classes. Each image is 32 by 32 with 3 channels for the RGB values. We normalize the images to have zero mean and unit variance. Additionally, for the ResNet models, we apply the same data-augmentation techniques as in the original ResNet paper (He et al., 2015): horizontal flipping and random cropping.

**CIFAR10 pixel-level**   In Long Range Arena (Tay et al., 2021), CIFAR10 images are flattened as sequences of 3-dimensional vectors. Sequence models such as RNNs can then be applied to image classification.

**ImageNet32**   The ImageNet32 (Chrabaszcz et al., 2017) dataset contains 1.3M images with 1000 classes. Each image is 32 by 32 with 3 channels for the RGB values. We process this the same way as CIFAR10.

**Wikitext103**   Wikitext103 is a dataset containing texts extracted from Wikipedia. We used two variants depending on the tokenizer used to convert the text into token indices: character-level (the 210 most common characters in the dataset) and word-level (a BPE tokenizer with 16k token, trained on the dataset as in GPT-2).

**MNLI**   The Multi-Genre Natural Language Inference dataset (Williams et al., 2018) is a task from the GLUE benchmark (Wang et al., 2018). It contains 433k sentence pairs, labelled with entailment information. The model has to predict whether the two sentences are an entailment, a contradiction, or neutral. For evaluation, we use the validation split with domains matching the training set.

## C.3   MODELS

**Pre-act ResNet**   We use the same architecture as the original ResNet for CIFAR10 (He et al., 2015), using pre-activations as in (He et al., 2016). We only modify the downsampling layers as described in Appendix G. We use:

$$g_i(x) = \text{Upsample}_i(\text{Block}_i(\text{Downsample}_i(x))) \qquad r_i(x, z) = x + z \qquad (14)$$

**ResNet**   Similarly, we use the same architecture as the original ResNet for CIFAR10 (He et al., 2015), and we only modify the downsampling layers as described in Appendix G. We use:

$$g_i(x) = \text{Upsample}_i(\text{Block}_i(\text{Downsample}_i(x))) \qquad r_i(x, z) = \text{ReLU}(x + z) \qquad (15)$$

**GPT-2**   We use a transformer model following the original GPT-2 architecture (Radford et al., 2019), with only smaller dimensions and vocabulary size. Note that GPT-2 uses pre-normalization, *i.e.* the layer-norm is applied at the beginning of all layers, and the residual connection is purely linear. This implies that the residual connection is linear:

$$g_i(x) = \text{Block}_i(x)) \qquad r_i(x, z) = x + z \qquad (16)$$

However, the above choice for $g_i$ and $r_i$ is not suited and makes Highway-BP diverge. We believe this is because the layers learn to cancel part of their residual connection. Taking this into account, in our experiments we used:

$$g_i(x) = \text{Block}_i(x)) + \gamma x \qquad r_i(x, z) = (1 - \gamma)x + z \qquad (17)$$

where we picked $\gamma = 0.2$ with minimal tuning. Finding better ways of choosing $\gamma$, understanding why it is necessary, and studying how this impacts Highway-BP, are future works that could help to considerably improve Highway-BP on such models.

Note that the two equations are mathematically equivalent when doing backpropagation, but the latter greatly improves the convergence of Highway-BP. In particular, $\gamma = 0$ is equivalent to the previous equation, and $\gamma = 1$ makes Highway-BP behave exactly like the fixed-point iteration baseline.

**RoBERTa**   We finetune the pre-trained RoBERTA-base model introduced by Liu et al. (2019). Compared to GPT-2, RoBERTA uses post-layernorm, *i.e.* applies a LayerNorm layer after each

Table 4: Hyperparameters used in the deep models experiments. In language modeling tasks, the last input dimension is the vocabulary size.

| Model | Pre-act ResNet32 | ResNet110 | ResNet56 | GPT-2 | RoBERTa |
|---|---|---|---|---|---|
| Dataset | CIFAR10 | CIFAR10 | ImageNet32 | Wikitext103 | MNLI |
| Optimizer | SGD | SGD | SGD | AdamW | AdamW |
| Epochs | 200 | 200 | 100 | 5.3 | 2 |
| Learning rate | 0.1 | 0.1 | 0.1 | 1e-3 | 3e-5 |
| Batch size | 128 | 128 | 128 | 128 | 32 |
| Warmup steps | 0 | 0 | 0 | 2k | 2.5k |
| Momentum | 0.9 | 0.9 | 0.9 | 0.9 | 0.9 |
| Adam $\beta_2$ | – | – | – | 0.98 | 0.999 |
| Weight decay | 1e-4 | 1e-4 | 1e-4 | 0.1 | 0.1 |
| Gradient clip | – | – | – | 1.0 | 1.0 |
| Input shape | (32, 32, 3) | (32, 32, 3) | (32, 32, 3) | (256, 16k) | (128, 50k) |
| Hidden dim | $16 \rightarrow 64$ | $16 \rightarrow 64$ | $16 \rightarrow 64$ | 256 | 768 |
| Layers | 32 | 110 | 56 | 12 | 12 |
| Params | 464k | 1.7M | 917k | 14.5M | 124M |

residual connection.

$$f_i(x) = \text{LN}_i(x + \text{Block}_i(x)) \tag{18}$$

$$= \frac{x + \text{Block}_i(x) - \overbrace{\mathbb{E}[x + \text{Block}_i(x)]}^{\mu_i(x)}}{\underbrace{\sqrt{\text{Var}(x + \text{Block}_i(x)) + \epsilon}}_{\sigma_i(x)}} \odot \alpha_i + \beta_i \tag{19}$$

$$= x \odot \underbrace{\frac{\alpha_i}{\sigma_i(x)}}_{a_i(x)} + \underbrace{(\text{Block}_i(x) - \mu_i(x)) \odot \frac{\alpha_i}{\sigma_i(x)} + \beta_i}_{b_i(x)} \tag{20}$$

$$= x \odot a_i(x) + b_i(x) \tag{21}$$

Which leads to the natural choice for $g_i$ and $r_i$:

$$g_i(x) = [a_i(x), b_i(x)] \qquad\qquad r_i(x, z) = x \odot z_1 + z_2 \tag{22}$$

However, similarly to the GPT-2 experiment with Equation 17, we introduce a small change to improve the convergence of Highway-BP:

$$g_i(x) = [(1 - \gamma)a_i(x), b_i(x) + \gamma x \odot a_i(x)] \qquad r_i(x, z) = x \odot z_1 + z_2 \tag{23}$$

This comes naturally when, instead of splitting $x + \text{Block}_i(x)$ into $x$ and $\text{Block}_i(x)$, we split it into $(1 - \gamma)x$ and $\gamma x + \text{Block}_i(x)$. We found $\gamma = 0.8$ to be a good choice, however again we believe there are still many things to understand with a lot of room for improvement, which we leave as future work.

**RNNs** The RNN models contain a linear layer to project the input to the hidden dimension, followed by the RNN layers, and then a classifier. The classifier is linear for language modeling (Wikitext103), and is a two-layer MLP for sequence classification (CIFAR10 pixel-level). In addition, when stacking multiple RNN layers, we introduce residual connections which greatly improve convergence speed.

## C.4 HYPERPARAMETERS

We report the hyperparameters used for sequential models in Table 4, and for RNNs in Table 5. Parameters that are not showed are set using their default values.

Table 5: Hyperparameters used in the RNN experiments. In language modeling tasks, the input dimension is the vocabulary size.

| RNN type | LSTM | GRU | GRU | GRU |
|---|---|---|---|---|
| Dataset | Wikitext103 char | Wikitext103 char | Wikitext103 | CIFAR10 |
| Training steps | 10k | 10k | 10k | 2.5k |
| Learning rate | 1e-3 | 1e-3 | 1e-3 | 1e-3 |
| Batch size | 128 | 128 | 128 | 128 |
| Warmup steps | 1k | 1k | 1k | 250 |
| Sequence length | 256 | 256 | 256 | 1024 |
| Input dim | 210 | 210 | 16k | 3 |
| Hidden dim | 512 | 512 | 512 | 64 |
| Layers | 1 | 1 | 3 | 1 |
| Params | 2.3M | 1.8M | 21.1M | 29.8k |

## D  MEMORY ANALYSIS

In this section we perform a simplified estimation of the memory usage of backpropagation and Highway-BP. Backpropagation requires the following memory for each layer:

$$\mathcal{M}_{\text{BP}} = \mathcal{M}_{\text{weights}} + \mathcal{M}_{\text{cache}} + 2\mathcal{M}_{h_i} \tag{24}$$

where $\mathcal{M}_{\text{weights}}$ is the size of the layer weights, $\mathcal{M}_{\text{cache}}$ is the space taken by the intermediate variables kept in memory for backpropagation, $\mathcal{M}_{h_i}$ is the size of the hidden state $h_i$ (and of its gradient).

A Highway-BP iteration involves a few variables per layer:

- $w_i^k$ (the current gradient estimate): it plays the same role as $\frac{\partial \mathcal{L}}{\partial h_i}$ in backpropagation, as it is backpropagated through the residual block $g_i$, so it does not take additional memory.

- $v_i^{k+1}$: similarly, plays the same role as $\frac{\partial \mathcal{L}}{\partial h_{i-1}}$ in backpropagation.

- $w_i^0$ (the initial estimate): this variable needs to be stored.

- $K_i$ (the Jacobian of the residual connection): already included in the intermediary variables stored by backpropagation.

- CumSumProd operation: we see from Algorithm 1 that it requires two tensors per layer if we use inplace operations. The first tensor is $P_i^{(m)}$: the product of the $K_i$ (if $K_i = I$ as in transformers for instance, there is nothing to store) which usually takes the same space as $h_i$ (*e.g.* if $K_i$ is diagonal). The second tensor is $u_i^{(m)}$, for which we can reuse the memory allocated to $v_i^{k+1}$ using inplace operations.

This leads to only two additional tensors to store ($w_i^0$ and $P_i^{(m)}$):

$$\mathcal{M}_{\text{Highway-BP}} = \mathcal{M}_{\text{weights}} + \mathcal{M}_{\text{cache}} + 4\mathcal{M}_{h_i} \tag{25}$$

Note that in practice $\mathcal{M}_{\text{cache}}$ is what takes most of the memory. In transformers for instance, $\mathcal{M}_{\text{cache}} \approx 17\mathcal{M}_{h_i}$. Nevertheless, there can be a non-negligible memory overhead for models with small residual blocks.

## E  PRACTICAL CONSIDERATIONS IN A DISTRIBUTED SETTING

While Highway-BP is a new way of speeding up training, it has to be noted that is also does not conflict with the standard techniques used in distributed training. The most widespread technique is data parallel, which splits the input bach across devices to process each part in parallel. This can still be used with Highway-BP, since our algorithm is only used for computing (or approximating) the gradient. In a layer parallel (or pipeline parallel) setting, layers are on different devices, which is exactly where Highway-BP would be useful.

The way Highway-BP may conflict with other distributed modes is if the number of devices is constrained. One would then have to decide which devices should be used either for increasing the batch size (data parallel) or speeding up backpropagation (Highway-BP).

Regarding the potential communication overhead, the main difference with backpropagation is the CumSumProd operation which performs a prefix scan of tensors stored on all devices (one per layer). While the algorithm is parallel, it is nontrivial to implement efficiently and there can be different strategies. With a fully distributed strategy, there will be more communications since a layer needs to send its tensor to two other layers at each iteration. Another approach is a centralized strategy, where all tensors are gathered together, CumSumProd is computed locally, and the results are sent back to the layers, which reduces the number of communications between devices. Finding out which implementation is better is an important future work. Note as well that there are multiple parallel prefix scan algorithms, each with different advantages.

## F  SPEED COMPARISON BETWEEN TRAINING ALGORITHMS FOR RNNS

Table 6: Computation times for different RNN tasks, algorithms, and $k$ values.

| Task | Algorithm | Training step time (ms) w.r.t. $k$ | | | | | | |
|------|-----------|------|-----|-----|-----|-----|-----|-----|
| | | **0** | **1** | **2** | **3** | **5** | **10** | **20** |
| **1 LSTM layer (L=256)** | Backpropagation | 261 | — | — | — | — | — | — |
| | Fixed-point iteration | 81 | 92 | 99 | 111 | 128 | 173 | 262 |
| | Highway-BP | 87 | 101 | 114 | 125 | 154 | 218 | 347 |
| **1 GRU layer (L=256)** | Backpropagation | 209 | — | — | — | — | — | — |
| | Fixed-point iteration | 62 | 69 | 73 | 79 | 92 | 119 | 176 |
| | Highway-BP | 66 | 77 | 88 | 97 | 118 | 166 | 266 |
| **3 GRU layers (L=256)** | Backpropagation | 687 | — | — | — | — | — | — |
| | Fixed-point iteration | 226 | 243 | 260 | 281 | 316 | 400 | 573 |
| | Highway-BP | 240 | 268 | 300 | 332 | 389 | 538 | 839 |
| **1 GRU layer (L=1024)** | Backpropagation | 715 | — | — | — | — | — | — |
| | Fixed-point iteration | 191 | 207 | 204 | 203 | 209 | 216 | 241 |
| | Highway-BP | 205 | 210 | 210 | 210 | 230 | 248 | 292 |

## G  HIGHWAY-BP AND DOWNSAMPLING LAYERS IN RESNETS

For Highway-BP to be effective on ResNet models, we need to adapt the downsampling layers from the original architecture. Indeed, they are not convenient to handle in our framework since the downsampling occurs in the residual connection, and the default downsampling layers have no simple way of computing and factorizing their Jacobians.

We take inspiration from $i$-RevNet (Jacobsen et al., 2018), in which the downsampling layers are invertible: the image is split into blocks of size $s \times s$, which are then flattened to produce one vector per block. This is already much easier to handle in Highway-BP, for instance in a pre-activation ResNet this means $K_i$ is simply a permutation. Furthermore, they are also easily factorizable, since downsampling with block size $s_1$ and then $s_2$ is equivalent to doing it once with block size $s_1 s_2$. However, a limitation is that when this layer divides the width and height by $s$, it also increases the vector dimension by $s^2$, as opposed to the original ResNet where the dimension increases by $s$. This leads to the number of parameters exploding.

The above downsampling layer can be modified to fix this issue. Before flattening the blocks, we perform an average over their height dimension. We thus obtain rows of $s$ vectors, which once flattened produce vectors scaled by $s$ instead of $s^2$. We used this modification in our implementation.

Note that the $i$-RevNet downsampling and ours are right-invertible, *i.e.* we can define an *Upsample* function such that Downsample(Upsample($x$)) = $x$. We take advantage of such properties in our implementation by wrapping all residual blocks by a downsample layer and an upsample layer, using

the same block sizes as the original ResNets, instead of placing the downsample layers in the residual connections. This way, the residual connections are even simpler, since we have $K_i = I$. For pre-activation ResNets, this is mathematically equivalent to the original implementation. For ResNets with ReLU between blocks, this is slightly different since $\text{Downsample}(\text{ReLU}(\text{Upsample}(x))) \neq x$, but has no noticeable impact.

