# OpenReview forum: "Accelerated training through iterative gradient propagation along the residual path"
_ICLR.cc/2025/Conference — ICLR 2025 Oral_

### Official Review · Reviewer_MEWH · 2024-10-15

**Soundness:** 3
**Presentation:** 3
**Contribution:** 4
**Rating:** 8
**Confidence:** 4

**Summary:**

This paper introduces a new algorithm for approximating backpropagation in deep sequential neural network models: Highway Backpropagation (Highway-BP)

Motivation: Backprop is computationally expensive, and has sequential dependence in depth making it hard to parallelize.
Key idea: Highway-BP leverages residual connections in modern architectures to decompose the gradient into paths of different lengths. It then iteratively computes an approximation of the gradient. Pruning some of the gradient paths enables Highway-BP to compute the approximate backprop in parallel. Doing this iteratively allows the approximate gradient to approach the true gradient over iterations.




I really like this paper. In my head I'm drawing a parallel with the "Residual Networks Behave Like Ensembles of Relatively Shallow Networks" paper. In Figure 1 from that paper, they "unravel" the residual connection of an N=3 block network.
When I read your paper, I immediately drew a parallel with that paper where, a view of your method, is that you only take backprop thru the right diagonal portion of the "unraveled" representation of Figure 1(b) of the "Residual Networks Behave Like Ensembles of Relatively Shallow Networks" paper.
Its brilliant!
(I am not an author of that paper :) )

**Strengths:**

(1) Flexibility: Can be applied to various architectures like ResNets, Transformers, and RNNs.

(1.5) Experiments: Tested on image classification (CIFAR10/100) and language modeling (Wikitext103) tasks. Compared against standard backpropagation and fixed-point iteration (for RNNs). Achieved comparable performance to backpropagation with fewer iterations (k ≤ 5 in most cases). Showed consistent speedups (2x-4x) over standard backpropagation.

(2) Paper presents a tradeoff frontier: Number of iterations (k) controls the trade-off between speed and accuracy

Edit:
(3) The parallelism unlocked by this method can greatly accelerate training speeds producing more efficient and faster training setups. The fact that it generalizes to a lot of settings makes this paper :chefs-kiss:
(see Appendix C for potential speedup wins (but this could be better presented in the main body))

**Weaknesses:**

This proposes a way to accelerate training at the cost of model performance
The true cost of AI is inference NOT training. I'd pay more for training if it produced a better model (and therefore made TCO cheaper)...

The results obviously show that the using the method produces worse models


I'll be the echo chamber (somebody has gotta do it): I want to see this applied to large scale autoregressive LLMs. Show me LLaMa training or I don't care!
\s

**Questions:**

The results obviously show that the using the method produces worse models.
Have you looked at methods for recovering performance?
Have you tried increasing k over training?
In figure 3, I can see a schedule in which, for the first 500 steps, you use k=1, then for the next 500 steps, you use k=2, then for the next 1000 steps, you use k=5, then eventually you use backprop a the end of training to fully recover performance.


Edit:
Can Figure 3 be redone but with training time on the x-axis (instead of time)? Appendix C shows that your method will look more favorable when this is done.

---

> ### Author Response · Authors · 2024-11-19
>
> > In my head I'm drawing a parallel with the "Residual Networks Behave Like Ensembles of Relatively Shallow Networks" paper.
> >
>
> Thank you for this valuable resource which we were not aware of, it perfectly describes the intuition we had when designing Highway-BP.
>
> > This proposes a way to accelerate training at the cost of model performance The true cost of AI is inference NOT training. I'd pay more for training if it produced a better model (and therefore made TCO cheaper)...
> The results obviously show that the using the method produces worse models. Have you looked at methods for recovering performance? Have you tried increasing k over training? In figure 3, I can see a schedule in which, for the first 500 steps, you use k=1, then for the next 500 steps, you use k=2, then for the next 1000 steps, you use k=5, then eventually you use backprop a the end of training to fully recover performance.
> >
>
> This is a very relevant point. While $k$ allows to control the speed vs training quality tradeoff, we still aim at producing models matching backpropagation. A first use would be to simply choose $k$ large enough to match backpropagation, in which case the model is as good as with backpropagation but was trained faster. We indeed showed that a low $k$ can be enough to match backpropagation exactly, so there is no loss in the model performance.
>
> Another use, which you describe and which we also had in mind, is the idea of increasing $k$ during training. At the beginning of training we observe that low values of $k$ are enough, but the gap increases as training goes on so we may need to increase $k$. We are already exploring this, and we are also trying to get theoretical insights on the choice of $k$. Ideally, we would also wand it to adapt $k$ dynamically and automatically.
>
> > I'll be the echo chamber (somebody has gotta do it): I want to see this applied to large scale autoregressive LLMs. Show me LLaMa training or I don't care!
> >
>
> We do want to evaluate Highway-BP on more extreme tasks such as LLM training. But this would require significant refactoring of our code and is planned in future work. The benefits of Highway-BP should be particularly visible in a large-scale distributed setting like the one used to train big LLMs.
>
> > Can Figure 3 be redone but with training time on the x-axis (instead of time)? Appendix C shows that your method will look more favorable when this is done.
> >
>
> Thank you for this suggestion, we will include such a figure in the final version of the paper.

---

> ### Author Response · Authors · 2024-11-29
>
> We have just updated the paper, and in particular made the following changes following your suggestions:
> - We have made the speedups more visible in the RNN experiments, by adding a summarized table of the speedups for all RNN experiments, and a plot of loss vs. runtime for the largest RNN model.
> - While this is not a Llama training, we have included a new experiment about fine-tuning a larger transformer, RoBERTa (124M parameters), on the MNLI task.
> - We have also included the "Residual Networks Behave Like Ensembles of Relatively Shallow Networks" paper as a valuable reference.

---

### Official Review · Reviewer_TXde · 2024-11-02

**Soundness:** 4
**Presentation:** 3
**Contribution:** 3
**Rating:** 8
**Confidence:** 4

**Summary:**

Summary: This paper proposes Highway Backpropagation, an algorithm for accelerating the training of neural nets with residual connections (such as ResNets, RNNs, and Transformers) by skipping some of the expensive backpropagation steps. In a nutshell, backpropagating a gradient through a layer consisting of a weight matrix plus a residual connection consists of a ‘cheap’ part (the residual) and an expensive part (the weight matrix), and the total gradient can be represented as the sum of all combinations of going through either. By omitting some of the terms of that summation - specifically those that have more than $k$ ‘expensive’ components, the gradient calculation can be sped up. The authors address the practical concerns of the implementation and provide experimental results validating their setup.

**Strengths:**

This paper tackles a crucially important problem, which is the enormous cumulative expense of training machine learning models, especially in the world of transformers. The idea seems to be quite original, and makes a great deal of intuitive sense. It is also straightforward to implement.

Overall, the main body of the paper is generally well-written and lays out its argument in a logical and even exciting way. The mathematical argument is very clearly presented. Further, the practical considerations of creating an implementation with real-world speedups are addressed.

The training dynamics section is nicely presented and interesting.

**Weaknesses:**

### Main weaknesses:

The main weakness of the paper is the experimental section. While the method is clever, the real value of the idea is in whether it is actually useful, and unfortunately the experimental section does not sufficiently cover this point. Of course, we cannot expect a finely optimized implementation in a research work. However, this reviewer doesn’t see a reason that a basic, workable implementation couldn’t have been built for standard architectures (e.g., ResNets) or something much closer to them. Because of the authors’ use of these nonstandard implementations, at least some of the actual numerical results are very far below what could be expected.

For instance, for CIFAR100, a standard ResNet50 architecture would have an accuracy of over 85%; the 50-layer architecture used in the paper (albeit with about a quarter of the total parameters of a RN50)  has an accuracy of 40%. A ResNet20, which has far fewer parameters, should still be able to get something like 70-75%. The authors do not address or justify this gap.

Likewise, real-world speedup numbers are not presented. Again, the point regarding the practical difficulty of creating a CUDA-optimal distributed setup is well taken, but the question remains unanswered. Perhaps it may have been possible to create a less-optimized standard implementation of backpropagation (e.g., on CPU) and compare this to the proposed one.

Finally, there is no guidance for selecting the right $k$ for a new task. This omission makes it difficult to use Highway Backpropagation in practice, even if an efficient implementation is available.

### Smaller issues:

The writing style of the introduction/related work sections is often vague or confusing.
* Like 36: what is meant by “transformers only defer the problem”?
* Line 41: What is meant by “often involve trade-offs between speed and task performance”? Aren’t these tradeoffs always involved?
* Line 42: What is meant by “leverage the recent layers”?
* Line 45: What is meant by “significantly improves optimization”? Can vague statements like this be made more precise?
* Line 51: What is meant by “derived from an original derivation”?
* Line 75: “where gradients can either diminish or grow uncontrollably” is completely redundant with line 73.
* Line 92: “While this seems attractive, the reality differs” - can this be made more clear and precise?
* Line 93: “computing the Jacobian matrix of all layers”: this is misleading, since the proposed Highway-BP algorithm also requires this computation.

Line 192 typo provie -> provide

Line 231 typo $K$ -> $k$

**Questions:**

* Why is the CIFAR-10 and CIFAR-100 model accuracy so low for the chosen architectures, or, conversely, why were architectures with such low accuracy chosen?

* Especially for image classification, why was this method not tested in the most standard setups, for instance Imagenet-1K on a standard ResNet50? Admittedly, in a standard RN50 architecture, the residual layer spans several weight matrices, but this seems like a straightforward extension of the proposed method. For the architectures chosen, why is the number of parameters much lower than that of the standard models (for instance, a traditional ResNet30 would have about 25M parameters, versus this paper’s 4.1M)? What is meant in line 317 by “standard ResNet”?

* In line 367, what is meant by “very reasonable performances”?

* Is there an advantage to varying $k$ during the training process?

* Why are the intermediate losses necessary for the theoretical framework to be effective?   Likewise, why is the proposed algorithm restricted to generative and classification tasks (liNE 153)?

---

> ### Author Response · Authors · 2024-11-19
> **(Part 1/2)**
>
> Thank you for this valuable feedback, we are taking this into account and will update the paper soon. We are currently working to strengthen our experimental section and provide more detailed discussion of the practical considerations of Highway-BP. We are also going to clarify the typos and vague sentences, thank you for taking the time to report them. Please find below answers to your questions.
>
> > The main weakness of the paper is the experimental section. While the method is clever, the real value of the idea is in whether it is actually useful, and unfortunately the experimental section does not sufficiently cover this point. Of course, we cannot expect a finely optimized implementation in a research work. However, this reviewer doesn’t see a reason that a basic, workable implementation couldn’t have been built for standard architectures (e.g., ResNets) or something much closer to them. Because of the authors’ use of these nonstandard implementations, at least some of the actual numerical results are very far below what could be expected.
> For instance, for CIFAR100, a standard ResNet50 architecture would have an accuracy of over 85\%; the 50-layer architecture used in the paper (albeit with about a quarter of the total parameters of a RN50) has an accuracy of 40\%. A ResNet20, which has far fewer parameters, should still be able to get something like 70-75\%. The authors do not address or justify this gap.
> >
>
> As you rightfully point out, the results of ResNet models from Figure 2 are far below the expected accuracy. We have fixed this and are currently rerunning these experiments. Our early experiments reach about 92\% accuracy on CIFAR10 (matching and even beating the original paper), with a model and training scheme almost identical to the original:
>
> | Original paper$\quad$ |   BP  |       |      | Highway-BP |       |       |       |
> |----------------|:-----:|:-----:|:----:|:----------:|:-----:|:-----:|:-----:|
> |                |       |  k=0  |   1  |      2     |   3   |   4   |   5   |
> | 92.49          | 92.75 | 82.38 | 89.0 |    91.00   | 92.68 | 92.82 | 92.64 |
>
> We had used non-standard implementations because we wanted more light-weight models for testing, and didn’t know how to handle downsampling with Highway-BP (downsampling layers are difficult to handle since their Jacobian is nontrivial). Now, the model is almost identical to a true ResNet. The only difference lies in the downsampling layers, which we had to modify a bit. We first took inspiration from i-RevNet, which uses an invertible downsampling layer (which is essentially a reshape, splitting the image into blocks which are then flattened to produce single vectors). However this leads to the dimension increasing quadratically, which we wanted to avoid. We slightly modified it to reduce the dimension by averaging over the height of the blocks before flattening them, which works very well in practice (as stated above we even beat the original model).
>
> > Likewise, real-world speedup numbers are not presented. Again, the point regarding the practical difficulty of creating a CUDA-optimal distributed setup is well taken, but the question remains unanswered. Perhaps it may have been possible to create a less-optimized standard implementation of backpropagation (e.g., on CPU) and compare this to the proposed one.
> >
>
> We do present speedup numbers for the RNN experiments (in the appendices). We will make these more visible in the paper. Our non-optimized implementation already leads to a significant speedup for RNNs (between $\times2$ and $\times4$), which are easier to handle as the same cell is repeated which makes Highway-BP efficient even on a single GPU. On the opposite, sequential models require a distributed setting to observe a speedup, which is much technical to deal with and is not supported in our current implementation.
>
> > Finally, there is no guidance for selecting the right $k$ for a new task. This omission makes it difficult to use Highway Backpropagation in practice, even if an efficient implementation is available.
> >
>
> We have observed through our experiments that $k=5$ seems like a good choice in general. But for now, $k$ indeed needs to be tuned (e.g. a grid search) in order to reach the desired speed vs accuracy tradeoff. We also were thinking of adaptive strategies to make $k$ evolve during training (as you mentioned below, more discussion there). This would remove the need of selecting the right value for $k$. Another idea would be to do iterations until the gradient estimate does not change anymore, akin to fixed-point algorithms. We left exploring these approaches as future work but we will include a discussion about this.

---

> ### Author Response · Authors · 2024-11-19
> **(Part 2/2)**
>
> > Line 93: “computing the Jacobian matrix of all layers”: this is misleading, since the proposed Highway-BP algorithm also requires this computation.
> >
>
> Highway-BP actually does not require the computation of the full Jacobian. This is a key difference with the similar methods from the literature. We take advantage of the fact that in residual models, the Jacobian of a layer is made of two parts: $J_i$ (Jacobian of the residual block, expensive) and $K_i$ (Jacobian of the residual connection, cheap, e.g. the identity or a diagonal matrix). Highway-BP only computes $K_i$, which is fast by design. $J_i$ is used like in standard backpropagation, through jacobian-vector products which do not require computing the Jacobian. We will make sure this is clarified in the paper.
>
> > Why is the CIFAR-10 and CIFAR-100 model accuracy so low for the chosen architectures, or, conversely, why were architectures with such low accuracy chosen?
> Especially for image classification, why was this method not tested in the most standard setups, for instance Imagenet-1K on a standard ResNet50? Admittedly, in a standard RN50 architecture, the residual layer spans several weight matrices, but this seems like a straightforward extension of the proposed method. For the architectures chosen, why is the number of parameters much lower than that of the standard models (for instance, a traditional ResNet30 would have about 25M parameters, versus this paper’s 4.1M)? What is meant in line 317 by “standard ResNet”?
> >
>
> We have solved these issues (see the answer to the first weakness for more details). The extension to ResNets is not that straightforward because of downsampling layers, which we have now successfully modified to use Highway-BP while keeping the original performance. In addition, the ResNet model on CIFAR-100 was suffering from a strong overfitting. Now the models have the exact same number of parameters than the original ResNets.
>
> > In line 367, what is meant by “very reasonable performances”?
> >
>
> Results show that $k=0$ (which is a very extreme choice) already leads to decent performance (82\% accuracy on cifar10 for instance in the table above, and similarly for the RNN results in the next section, for example a perplexity of 50 on wikitext103). We agree that “very reasonable” is a bit too much, we will tone it down. Note that in the RNN section, the fixed-point iteration baseline with $k=0$ is worse than Highway-BP, showing that our initial gradient approximation $w_i^0$ is already a better guess and contributes to the better results with $k=0$.
>
> > Is there an advantage to varying $k$ during the training process?
> >
>
> Yes, this is indeed something we are very interested in. From the training curves, we observe that low values of $k$ are especially good at the beginning of the training, while the gap increases when training continues. A natural strategy would be to increase $k$, or to adapt it dynamically to adjust the quality of the approximation for instance. Another idea would be to do iterations until the gradient estimate does not change anymore, akin to fixed-point algorithms. We left exploring these approaches as future work but we will include a discussion about this. We believe this would be a good way of using Highway-BP to accelerate training without losing any final model performance.
>
> > Why are the intermediate losses necessary for the theoretical framework to be effective? Likewise, why is the proposed algorithm restricted to generative and classification tasks (liNE 153)?
> >
>
> The intermediate losses are not necessary for Highway-BP to work. This is a key difference with the “fixed-point iteration” baseline we use for RNNs, which needs them. We include these losses in the framework so that it can handle them it they exist, otherwise they can be set to 0.
>
> In practice, intermediate losses are only present for RNNs when generating sequences (as in our language modeling tasks). The other tasks are classification tasks and only involve a final loss using the last hidden state $h_L$.
>
> This is what we mean by “generative and classification tasks”, although the phrasing is not clear (we will reformulate that part). This is by no means a restriction, on the contrary the framework can actually handle more diverse tasks, with any loss functions that involve any hidden state from $h_1$ to $h_L$.

---

> > ### Comment · Reviewer_TXde · 2024-11-20
> > **Thank you for the rebuttal**
> >
> > I thank the authors for their very thorough rebuttal. My concerns were addressed, and I will update my scores accordingly.

---

> ### Author Response · Authors · 2024-11-29
>
> Following our discussion, we have just updated the paper, and in particular made the following changes:
> - As mentioned in our previous answer, the ResNet experiments are now using the same architecture and number of parameters as the original paper, and have the same performance (even beating it). For instance, the ResNet110 now reaches 93.8% accuracy on CIFAR10 with both backpropagation and Highway-BP with $k \geq 4$.
> - We have made the speedups more visible in the RNN experiments, by adding a summarized table of the speedups for all RNN experiments, and a plot of loss vs. runtime for the largest RNN model.
> - We are thankful for taking the time to report typos and unclear sentences, we have fixed them.

---

### Official Review · Reviewer_rbdV · 2024-11-04

**Soundness:** 3
**Presentation:** 3
**Contribution:** 3
**Rating:** 8
**Confidence:** 3

**Summary:**

This paper considers gradient computation in network architectures with multiple residual connections. The authors propose the Highway-BP method to improve the efficiency of gradient computation/estimation by decoupling and rearranging multiple computational paths in vanilla BP. The proposed algorithm leverages the fact that computational burdens differ between the two paths within each block and iteratively approximates the gradient from easier to more challenging paths, allowing trade-offs between accuracy and efficiency through adjustable iteration steps. Numerical results indicate that the approximate gradient, with a certain number of iterations, achieves comparable results to the true gradient, demonstrating its potential applicability.

**Strengths:**

The authors present an interesting idea for improving the sequential computation in BP, utilizing the shared structural characteristics of mainstream models, which may facilitate further refined explorations in the future. The experimental results in the paper are comprehensive, demonstrating potential advantages in terms of efficiency.

**Weaknesses:**

My main concern is that unrolling the recursive computation in vanilla BP requires storing a significant number of additional variables, thereby increasing the algorithm’s storage demands. The authors do not appear to provide test results for memory consumption, which is an important consideration when developing algorithms for large-scale models. From the results in Table 3, the proposed algorithm shows a higher time overhead than the baseline, and with only 20 iterations, the time consumption is nearly equivalent to that of vanilla BP, indirectly suggesting that the additional read/write overhead may not be insignificant.

The efficiency of the proposed algorithm lacks theoretical guarantees, and the experimental results have minor flaws. Please see the question part for details. The comparison includes only the fixed-point iteration algorithm. Algorithms mentioned in the literature and other approximation-based acceleration methods for BP are not included.

**Questions:**

- The authors should provide a more detailed experimental setup in the paper. In Figure 2, the final performance of vanilla BP appears to differ significantly from its usual training outcomes. Might this be due to insufficient training? As observed in Figure 3, the effectiveness of Highway-BP's approximation seems limited to the relatively easy early stages of training, suggesting that the accuracy results in Figure 2 may need to be presented at multiple time points.

- The paper includes empirical tests of the algorithm, but could the authors provide a theoretical analysis of how the approximation error varies with $k$? Alternatively, could a more rigorous boundary be identified for the types of network structures where this approximation applies? Additionally, conducting a comprehensive analysis of the time and space complexity of Highway-BP is also worthwhile and important.

- Since the paper aims to alleviate the sequentiality in BP, which is a challenge that typically arises in large models, could the authors evaluate whether the complex computation of Highway-BP might hinder the application of other crucial techniques in this area, such as data parallelism or distributed training? Furthermore, how might these structural changes to gradient computation impact communication and read/write overhead?

---

> ### Author Response · Authors · 2024-11-19
> **(Part 1/2)**
>
> Thank you for this valuable feedback, we are taking this into account and will update the paper soon. We are currently working to strengthen our experimental section and provide more detailed discussion of the practical considerations of Highway-BP. Please find below answers to your questions.
>
> > My main concern is that unrolling the recursive computation in vanilla BP requires storing a significant number of additional variables, thereby increasing the algorithm’s storage demands. The authors do not appear to provide test results for memory consumption, which is an important consideration when developing algorithms for large-scale models.
> >
>
> Backpropagation also unrolls the recursive computation, since during the forward pass it needs to store almost all hidden states in order to backpropagate gradients later. This means that the memory scales with the number of layers. In Highway-BP, we actually need two more tensors per layer (the same shape as the hidden states), which is not a big overhead in comparison. We are including a detailed discussion about the memory overhead of Highway-BP in the paper.
>
> > From the results in Table 3, the proposed algorithm shows a higher time overhead than the baseline, and with only 20 iterations, the time consumption is nearly equivalent to that of vanilla BP, indirectly suggesting that the additional read/write overhead may not be insignificant.
> >
>
> The strength of our method (especially compared to the baseline) is that a low $k$ is enough. We show in our experiments that a value around $k=5$ already produces very good approximations of the gradient. We also believe that our implementation could be further optimized — here we focused mainly on flexibility and reliability.
>
> The difference between the baseline and Highway-BP is the CumSumProd operation, which indeed seems to take a non-negligible part of the computation time. We will reformulate Eq. 12 to make this term clearer.
>
> Note that very low values of $k$ can be especially useful at the beginning of training. We believe that $k$ could be increased over time to speed up at least part of the training, ending with a true backpropagation.
>
> > The comparison includes only the fixed-point iteration algorithm. Algorithms mentioned in the literature and other approximation-based acceleration methods for BP are not included.
> >
>
> The issue with most other methods is that they are often not optimized, and neither is our implementation, which makes it hard to compare them with Highway-BP. Still, we are considering including the closest method: Günther et al., “Layer-Parallel Training of Deep Residual Neural Networks”, 2019. They use the MGRIT solver to approximate the gradient in parallel and iteratively, similarly to Highway-BP. They however did not scale their experiments any further than MNIST.

---

> ### Author Response · Authors · 2024-11-19
> **(Part 2/2)**
>
> > The authors should provide a more detailed experimental setup in the paper. In Figure 2, the final performance of vanilla BP appears to differ significantly from its usual training outcomes. Might this be due to insufficient training? As observed in Figure 3, the effectiveness of Highway-BP's approximation seems limited to the relatively easy early stages of training, suggesting that the accuracy results in Figure 2 may need to be presented at multiple time points.
> >
>
> All the missing experimental details are going to be added in the appendices.
>
> As you rightfully point out, the results of ResNet models from Figure 2 are far below the expected accuracy. We have fixed this and are currently rerunning these experiments. Our early experiments reach about 92\% accuracy, with a model and training scheme much closer to the original:
>
>
> | Original paper$\quad$ |   BP  |       |      | Highway-BP |       |       |       |
> |----------------|:-----:|:-----:|:----:|:----------:|:-----:|:-----:|:-----:|
> |                |       |  k=0  |   1  |      2     |   3   |   4   |   5   |
> | 92.49          | 92.75 | 82.38 | 89.0 |    91.00   | 92.68 | 92.82 | 92.64 |
>
> Insufficient training was one issue, but also the architecture itself was lacking in multiple ways. We had used non-standard implementations because we wanted more light-weight models for testing, and didn’t know how to handle downsampling with Highway-BP (downsampling layers are difficult to handle since their Jacobian is nontrivial). Now, the model is almost identical to a true ResNet. The only difference lies in the downsampling layers, which we had to modify a bit. We first took inspiration from i-RevNet, which uses an invertible downsampling layer (which is essentially a reshape, splitting the image into blocks which are then flattened to produce single vectors). However this leads to the dimension increasing quadratically, which we wanted to avoid. We slightly modified it to reduce the dimension by averaging over the height of the blocks before flattening them, which works very well in practice (as stated above we even beat the original model).
>
> > The paper includes empirical tests of the algorithm, but could the authors provide a theoretical analysis of how the approximation error varies with $k$? Alternatively, could a more rigorous boundary be identified for the types of network structures where this approximation applies? Additionally, conducting a comprehensive analysis of the time and space complexity of Highway-BP is also worthwhile and important.
> >
>
> We have relatively successful attempts at estimating the variance of the approximation error for a given $k$, and we may include this as an appendix. However this is hard to analyze given that the approximation quality highly depends on the properties of the Jacobians, which are unknown and evolve over training. We will also include more discussion about the time and space complexity.
>
> > Since the paper aims to alleviate the sequentiality in BP, which is a challenge that typically arises in large models, could the authors evaluate whether the complex computation of Highway-BP might hinder the application of other crucial techniques in this area, such as data parallelism or distributed training?
> >
>
> This is a very relevant point since the benefits of Highway-BP arise in a distributed setting.
>
> - Data parallel and distributed data parallel are a way of splitting the input batch to be processed by different devices, each computing the gradient, and then gathering and summing the gradients. This does not conflict with Highway-BP, since it is used for computing (or approximating) the gradient.
> - In a layer parallel (or pipeline parallel) setting, layers are on different devices, which is exactly where Highway-BP would be useful.
>
> The way Highway-BP may conflict with other distributed modes is if the number of devices is constrained. One would then have to decide which devices should be used either for increasing the batch size (data parallel) or speeding up backpropagation (Highway-BP).
>
> > Furthermore, how might these structural changes to gradient computation impact communication and read/write overhead?
> >
>
> The main difference with backpropagation is the CumSumProd operation which performs a prefix scan of tensors stored on all devices (=layers). While the algorithm is parallel it is nontrivial to implement efficiently and there can be different strategies. With a fully distributed strategy, there will be more communications since a layer needs to send its tensor to two other layers each iteration. Another approach is a centralized strategy, where all tensors are gathered together, CumSumProd is computed, and the results are sent back to the layers, which reduces the number of communications between devices.

---

> ### Author Response · Authors · 2024-11-29
>
> Following our discussion, we have just updated the paper, and in particular made the following changes:
> - As mentioned in our previous answer, the ResNet experiments are now using the same architecture and number of parameters as the original paper, and have the same performance (even beating it). For instance, the ResNet110 now has 1.7M parameters and reaches 93.8% accuracy on CIFAR10 with both backpropagation and Highway-BP for $k \geq 4$.
> - We have added a new experiment on RoBERTa, fine-tuning the model on the MNLI dataset. This is a good test for Highway-BP on larger models, as the model has 124M parameters.
> - We have included some discussions about the memory overhead, and practical considerations of using Highway-BP in a distributed setting.
> - All the experimental setups are now detailed in the appendices.
>
> Please let us know if you have any further questions or comments.

---

> > ### Comment · Reviewer_rbdV · 2024-12-01
> > **Response to Rebuttal**
> >
> > I thank the authors for their thoughtful response to my concerns. Most of these have been addressed, while I understand the difficulty and potential for improvement of the others after reviewing the authors' explanation. Therefore, I will raise my score accordingly.

---

### Official Review · Reviewer_vqXj · 2024-11-04

**Soundness:** 3
**Presentation:** 3
**Contribution:** 3
**Rating:** 6
**Confidence:** 4

**Summary:**

The paper introduces Highway backpropagation (Highway-BP), an algorithm that speeds up training of deep neural networks by approximating traditional backpropagation using parallelizable iterative methods. It leverages residual connections to propagate gradients efficiently and is adaptable to various architectures like ResNets and Transformers. Empirical studies show that Highway-BP achieves significant speedups with minimal performance loss, making it a promising method for accelerating deep learning model training.

**Strengths:**

Parallelization: The algorithm is designed to be parallelizable, which can significantly reduce training time, especially for very deep models or large-scale problems.
Architecture Flexibility: Highway-BP is adaptable to a variety of common neural network architectures, making it a versatile tool for different applications.
Theoretical Foundation: The paper is grounded in mathematical theorems that provide a theoretical foundation for the proposed method.

**Weaknesses:**

Generalization to All Models: Although Highway-BP is adaptable to many architectures, there may be specific models or scenarios where it does not perform as well as traditional backpropagation or other optimization methods. For example, experiments on Graph neural network, spiking neural network and mamba.
Hyperparameter Tuning: The algorithm introduces a new hyperparameter (the number of iterations, k), which requires tuning and may lead to different optimal values depending on the model and task. This can add complexity to the training process. A ablation on how to tune the parameter should be included.
Missing baselines: The paper compares Highway-BP with backpropagation and fixed-point iteration but may not fully address how it stacks up against other state-of-the-art optimization techniques. Here are some papers authors should include in their baseline:
Huang, Kai, et al. "Towards Green AI in Fine-tuning Large Language Models via Adaptive Backpropagation." arXiv preprint arXiv:2309.13192 (2023).
Wang, Ziteng, Jianfei Chen, and Jun Zhu. "Efficient Backpropagation with Variance-Controlled Adaptive Sampling." arXiv preprint arXiv:2402.17227 (2024).
Yang, Yuedong, et al. "Efficient low-rank backpropagation for vision transformer adaptation." Advances in Neural Information Processing Systems 36 (2024).
Scalability to Distributed Settings: The paper mentions the potential for distributed training but does not provide empirical results or a detailed discussion on how Highway-BP would perform in a distributed setting.

**Questions:**

How does Highway-BP compare with other advanced optimization techniques, especially in terms of convergence speed and final model accuracy?
What are the performance implications of Highway-BP when scaling to models with billions of parameters, and how does it handle memory and computational constraints?
Can the authors provide more theoretical analysis or proofs regarding the convergence properties of Highway-BP compared to traditional backpropagation? For example, give a lower bound for the improvement.

---

> ### Author Response · Authors · 2024-11-19
> **(Part 1/2)**
>
> Thank you for this valuable feedback, we are taking this into account and will update the paper soon. We are currently working to strengthen our experimental section and provide more detailed discussion of the practical considerations of Highway-BP. Please find below answers to your questions.
>
> > Generalization to All Models: Although Highway-BP is adaptable to many architectures, there may be specific models or scenarios where it does not perform as well as traditional backpropagation or other optimization methods. For example, experiments on Graph neural network, spiking neural network and mamba.
> >
>
> Indeed, since Highway-BP computes an approximation of the gradient, this approximation may be of higher or lower quality in different settings. Our experiments show promising results on diverse models and tasks, suggesting a good adaptability. But many more remain to be tested, as you point out. Still, Mamba in particular is likely to work similarly to transformers, since they share a lot in common.
>
> We are currently working on strengthening our experiments. We are not particularly familiar with graph neural networks, do you have a model / dataset in mind that we could evaluate Highway-BP on?
>
> > Hyperparameter Tuning: The algorithm introduces a new hyperparameter (the number of iterations, k), which requires tuning and may lead to different optimal values depending on the model and task. This can add complexity to the training process. A ablation on how to tune the parameter should be included.
> >
>
> We have observed through our experiments that $k=5$ seems like a good choice in general. But for now, $k$ indeed needs to be tuned (e.g. a grid search) in order to reach the desired speed vs accuracy tradeoff. We also were thinking of adaptive strategies to make $k$ evolve during training. For instance it could be set to $k=0$ at the beginning, and then increase, to finally finish the training with an exact backpropagation. Another idea would be to do iterations until the gradient estimate does not change anymore, akin to fixed-point algorithms. We left exploring these approaches as future work but we will include a discussion about this.
>
> > Missing baselines: The paper compares Highway-BP with backpropagation and fixed-point iteration but may not fully address how it stacks up against other state-of-the-art optimization techniques. Here are some papers authors should include in their baseline: Huang, Kai, et al. "Towards Green AI in Fine-tuning Large Language Models via Adaptive Backpropagation." arXiv preprint arXiv:2309.13192 (2023). Wang, Ziteng, Jianfei Chen, and Jun Zhu. "Efficient Backpropagation with Variance-Controlled Adaptive Sampling." arXiv preprint arXiv:2402.17227 (2024). Yang, Yuedong, et al. "Efficient low-rank backpropagation for vision transformer adaptation." Advances in Neural Information Processing Systems 36 (2024).
> >
>
> Thank you for pointing out these works to us. After investigation, it appears that two of them are orthogonal to our work as they can be combined with Highway-BP:
>
> - “Towards Green AI in Fine-tuning Large Language Models via Adaptive Backpropagation”: it can actually be combined with our method, as it is a way of selecting which layers will be updated, and our method computes the gradient for these layers
> - “Efficient low-rank backpropagation for vision transformer adaptation”: the paper focuses on accelerating backpropagation through each individual linear layer using a low-rank approximation. Highway-BP still requires backpropagating through linear layers, so the methods are orthogonal.
>
> The third paper, “Efficient Backpropagation with Variance-Controlled Adaptive Sampling”, is about pruning samples during backpropagation based on the norm of the gradient at the previous layer. It directly relies on the sequentiality of backpropagation which Highway-BP removes. An issue is that while we could evaluate it on our tasks, we could not be able to compare it to Highway-BP as both implementations are not optimized (they only report the theoretical speedup in FLOPs). In addition, this work is not parallelizable (layer-wise), so even is it were to provide better approximations, it still has a fundamental bottleneck compared to Highway-BP. We can however provide a more detailed complexity analysis of these baselines to compare them with Highway-BP if you find it relevant.
>
> Still, we are considering including a baseline that is very close to Highway-BP: Günther et al., “Layer-Parallel Training of Deep Residual Neural Networks”, 2019. They use the MGRIT solver to approximate the gradient in parallel, similarly to Highway-BP. They however did not scale their experiments any further than MNIST.

---

> ### Author Response · Authors · 2024-11-19
> **(Part 2/2)**
>
> > Scalability to Distributed Settings: The paper mentions the potential for distributed training but does not provide empirical results or a detailed discussion on how Highway-BP would perform in a distributed setting.
> >
>
> We indeed design Highway-BP to be parallelizable along the depth dimension. We provide very encouraging results for RNNs, with speedups between x2 and x4. The implementation for RNNs is very straightforward as the same cell is repeated, which leads to nice improvements on a single GPU. For sequential models however, our implementation is not optimized for a multi-GPU distributed setting. The goal of our paper is above all to analyze the performance of Highway-PB in terms of training quality. Still, we will include a detailed analysis of time, memory usage and communication in a distributed setting.
>
> > How does Highway-BP compare with other advanced optimization techniques, especially in terms of convergence speed and final model accuracy?
> >
>
> With $k \geq 5$, we observe that Highway-BP trains the model exactly like backpropagation for most of the tasks. This means that the convergence speed and final accuracy are the same in this case. When $k$ is too low however, the training quality is worse and we see a gap with backpropagation. Interestingly, while the final accuracy is worse, the convergence speed is usually the same — but it converges to a worse loss. Note that the training quality (i.e. convergence speed and final accuracy) can be controlled using $k$ to reach the desired tradeoff.
>
> > What are the performance implications of Highway-BP when scaling to models with billions of parameters, and how does it handle memory and computational constraints?
> >
>
> Highway-BP was designed with this application in mind, so that all GPUs can run in parallel to compute approximated gradients (and thus avoiding the backward-locking). While it does not require much more memory than backpropagation (about two more tensors per layer, with the same shape as the activations), what differs is the CumSumProd phase which may require special care to avoid too many communications between devices. We are including a detailed analysis of the time and memory implications of Highway-BP compared to backpropagation.
>
> > Can the authors provide more theoretical analysis or proofs regarding the convergence properties of Highway-BP compared to traditional backpropagation? For example, give a lower bound for the improvement.
> >
>
> We have tried to get a more theoretical analysis of Highway-BP, but did not obtain any practical result. It is possible to approximate the variance of the approximation error for a given $k$, which we may include as an appendix. The training dynamics are very complex, since the approximation error depends on the weights, which evolve during training. Still, we believe that a better theoretical understanding could be very useful for improving the method, for instance by guiding the choice of $k$.

---

> > ### Comment · Reviewer_vqXj · 2024-11-30
> > **Response to rebuttal**
> >
> > Thank you for your response. While I would still appreciate a more thorough theoretical evaluation of the proposed method, I understand the workload involved in that. The authors have addressed most of my concerns, and I will adjust my scores accordingly.

---

> ### Author Response · Authors · 2024-11-29
>
> Following our discussion, we have just updated the paper, and in particular made the following changes:
> - We have added a new experiment on RoBERTa, fine-tuning the model on the MNLI dataset. This is a good test for Highway-BP on larger models, as the model has 124M parameters. This experiment supports the claim that Highway-BP can be used in many architectures.
> - We have included discussions about the memory overhead, and practical considerations of using Highway-BP in a distributed setting.
>
> Please let us know if you have any further questions or comments.

---

### Official Review · Reviewer_3hir · 2024-11-05

**Soundness:** 3
**Presentation:** 3
**Contribution:** 3
**Rating:** 6
**Confidence:** 4

**Summary:**

This paper proposes a method to accelerate the backward pass of backpropagation through parallelization of the gradient calculation in deep neural networks with residual-like architecture. This is done by decomposing the gradient into a sum of terms, where the k-th term is the sum of all gradients passing through at most k blocks which are not a residual (i.e. skip-like) connection. By parallelizing this computation, an acceleration is achieved as long as the maximal k we compute is not too large (i.e., we need to drop values of k above some threshold). It is shown empirically that k=5 is approximately enough for good accuracy on several (small) benchmarks, while still achieving significant acceleration.

**Strengths:**

1. The problem of accelerating Backprop is important.

2. The idea is novel, original, and quite interesting.

3. The empirical results are promising.

4. The presentation is mostly clear.

**Weaknesses:**

1. The main issue is the scale of the experiments, which is rather small (e.g., no ImageNet). It is not clear to me that a small k would be enough to get both high accuracy and acceleration on more complicated problems than those shown in this paper.

2. The acceleration is only relevant for the backward pass, which is roughly a third of the total time of Backprop (which includes the forward pass, backward pass, and parameter gradient) and so this limits the overall benefit of this method.

**Questions:**

3. I think there are some small mistakes in the math. For example, in eq. 8:

$\quad$ a) The transition to eq. 8 from eq. 6 is not clear. Specifically, why did $w_{i}^{0}$ disappear?

$\quad$ b) The right side of eq. 8 seems different from the middle part of the equation, since $v_{i}^{k+1}$ is multiplied by $K_{i+1}$ in the middle part of the equation, but not in the right side of the equation.

Can the authors please check these?

---

> ### Author Response · Authors · 2024-11-19
>
> Thank you for this valuable feedback, we are taking this into account and are currently working to strengthen our experimental section. Please find below answers to your questions.
>
> > 1. The main issue is the scale of the experiments, which is rather small (e.g., no ImageNet). It is not clear to me that a small k would be enough to get both high accuracy and acceleration on more complicated problems than those shown in this paper.
> >
>
> This is indeed something we want to improve, and we are currently working to provide more meaningful experiments. In particular regarding the three ResNet experiments, we have changed the setting to make them more convincing by matching the SOTA results ($\approx$93\%) of the original ResNet paper for CIFAR-10 (instead of the $\approx$70\% before) -- see table below:
>
> | Original paper$\quad$ |   BP  |       |      | Highway-BP |       |       |       |
> |----------------|:-----:|:-----:|:----:|:----------:|:-----:|:-----:|:-----:|
> |                |       |  k=0  |   1  |      2     |   3   |   4   |   5   |
> | 92.49          | 92.75 | 82.38 | 89.0 |    91.00   | 92.68 | 92.82 | 92.64 |
>
> In addition, reviewer MEWH talked about the paper "Residual Networks Behave Like Ensembles of
> Relatively Shallow Networks", which exactly shows how a small $k$ is enough even for ResNets with 200 layers on ImageNet. Still, while CIFAR-10 and 100 are small image dataset, the Wikitext103 dataset we used for the language modeling experiments is much bigger (more than 100 million tokens) and we are considering scaling further the model.
>
> Note that even in a setting where $k$ is too low and noticeably deteriorates training quality, it may still be useful at the beginning of training. We observed that low values of $k$ are especially good at the start. In fact, we believe that making $k$ increase during training would provide the acceleration of Highway-BP, while matching the accuracy obtained with backpropagation. This is something we would like to explore.
>
> > 2.The acceleration is only relevant for the backward pass, which is roughly a third of the total time of Backprop (which includes the forward pass, backward pass, and parameter gradient) and so this limits the overall benefit of this method.
> >
>
> Indeed, we only accelerate the backward pass, which is one of the three main steps of the training algorithm as you pointed out. However, the backward pass is the most expensive out of them. For instance in https://arxiv.org/pdf/2006.15704, Figure 6, the backward pass takes about 75\% of the total time, with the forward pass taking about 25\% and the parameter update being almost negligible. Similarly, we report our speedups for the RNN experiments in the appendices, showing that we can get considerable speedups (x2 to x4 using a low $k$), which validates that the backward pass is the most time consuming part.
>
> > 3. I think there are some small mistakes in the math. For example, in eq. 8:
> >
> >
> > a) The transition to eq. 8 from eq. 6 is not clear. Specifically, why did $w_i^0$ disappear?
> >
> > b) The right side of eq. 8 seems different from the middle part of the equation, since $v_i^{k+1}$ is multiplied by $K_{i+1}$ in the middle part of the equation, but not in the right side of the equation.
> >
> > Can the authors please check these?
> >
>
> The equations are correct, but the notations are indeed misleading and unclear:
>
> a) We defined $u_i^k$ in Eq. 8 such that: $w_i^{k+1} = w_i^0 + u_i^{k+1}$. This notation is necessary to write the recursive relation $u_i^{k+1} = v_i^{k+1} + u_{i+1}^{k+1} K_{i+1}$, which is then solved using a prefix scan algorithm. $w_i^0$ is then added at the end to get $w_i^{k+1}$. We will clarify this in the paper.
>
> b) We found more natural to write $K_j K_{j-1} \dots K_{i+1}$ instead of a product with $\prod$ since the indices are reversed ($j \geq i$). It represents the reversed product of the $K_n$ for $n \in [i+1, j]$. As such, we have $v_{i+1}^{k+1}$ multiplied by $K_{i+1}$, while $v_i^{k+1}$ is multiplied by nothing since the interval $[i+1, i]$ is empty.

---

> ### Comment · Reviewer_3hir · 2024-11-24
> **Response to rebuttal**
>
> I thank the author for their response.
>
> 1. Can the authors please explain what is the issue preventing them from running an ImageNet experiment?
> 2. Please correct me if I'm missing something, but I think the authors and the cited paper both put the calculation of the weight gradient ($\partial L / \partial w$) inside the backward phase. However, this step (which I considered as the third phase (as it contains another matrix multiplication), not the optimizer update) is not accelerated by the current method --- which only accelerates the calculation of the neural gradients, i.e. the gradients concerning the neural layer $\partial L / \partial h$. Since the authors did get a significant acceleration in RNN experiments, are the authors saying that the cost of weight gradient calculation is also negligible?
> 3. I see, thanks for the clarification.

---

> ### Author Response · Authors · 2024-11-29
>
> > 1. Can the authors please explain what is the issue preventing them from running an ImageNet experiment?
> >
>
> The issue, which we have now resolved, was that the "ResNet" models we were using were not the same as the original ResNet models, and did not have any downsampling layers. We first focused on fixing this issue, and then we ran the experiments on CIFAR-10. ImageNet experiments are planned for the future since they require much more computational time and could not be done before the rebuttal deadline. We are considering upgrading one of the ResNet experiments to ImageNet for the final version of the paper. Also note that the paper "Residual Networks Behave Like Ensembles of Relatively Shallow Networks" provides evidence that Highway-BP should work well on very deep ResNets on ImageNet.
>
> In the meantime, we updated the paper to include a new experiment: fine-tuning a RoBERTa model on MNLI, a sequence classification task with 433k training samples. The model has 124M parameters, and hence is a good test for Highway-BP on larger models.
>
> > 2. Please correct me if I'm missing something, but I think the authors and the cited paper both put the calculation of the weight gradient ($\partial L / \partial w$) inside the backward phase. However, this step (which I considered as the third phase (as it contains another matrix multiplication), not the optimizer update) is not accelerated by the current method --- which only accelerates the calculation of the neural gradients, i.e. the gradients concerning the neural layer $\partial L/\partial h$. Since the authors did get a significant acceleration in RNN experiments, are the authors saying that the cost of weight gradient calculation is also negligible?
> >
>
> The part that is the most costly for backpropagation is the sequentiality of the algorithm, which is needed to compute $\partial L / \partial h_i$ for each layer. As you say, the computation of $\partial L / \partial w_i$ can be done afterwards.
>
> A key point here is that once we have $\partial L / \partial h_i$, $\partial L / \partial w_i$ can be computed independently, in parallel. This is what we do in Highway-BP, where after computing all the $\partial L / \partial h_i$, we do one more step to get the gradients w.r.t. the weights. Because it is parallelizable, the computation is very fast, especially compared to the sequential forward pass.
>
> As for standard backpropagation, auto-differentiation libraries like PyTorch and TensorFlow compute everything in one pass. However $\partial L / \partial h_i$ and $\partial L / \partial w_i$ are still computed in parallel on the GPU, and the sequentiality of the algorithm is still the bottleneck.
>
> So as you say, in both Highway-BP and backpropagation, the cost of computing $\partial L / \partial w_i$ is much lower than the forward and backward passes. We will make sure this is clear in the final version of the paper.

---

> > ### Comment · Reviewer_3hir · 2024-12-02
> > **Response**
> >
> > Thanks for the clarification and additional experiments. I do think an ImageNet experiment would help the overall quality of the paper. One last thing: is there any reason why are the acceleration measurements (table 6) done only on RNNs? Do you get similar accelerations with other architectures?

---

> > > ### Author Response · Authors · 2024-12-02
> > >
> > > > Thanks for the clarification and additional experiments. I do think an ImageNet experiment would help the overall quality of the paper.
> > > >
> > >
> > > We are currently running the experiment, but we will only be able to include the results after the discussion deadline.
> > >
> > > > is there any reason why are the acceleration measurements (table 6) done only on RNNs? Do you get similar accelerations with other architectures?
> > > >
> > >
> > > This is a valid point. Initially, we only wanted to focus on showing the performance of the algorithm in terms of the iterations $k$ vs accuracy tradeoff. The theoretical acceleration, as described in Equation 13, shows promising speedups for low values of $k$, for any architecture. However, for sequential models (e.g. ResNets, transformers), the parallelization is done by training in a distributed setting (multiple GPUs), which gets quite technical. We chose to leave an efficient implementation for future work, prioritizing the evaluation of whether the algorithm is practical for use, verifying the assumption that low values of $k$ suffice across various tasks and models.
> > >
> > > Still, because RNNs are essentially the same cell repeated $L$ times (instead of $L$ different layers in sequential models), they can be accelerated easily even on a single GPU. With our current implementation, this leads to the acceleration described in the paper.

---

### Author Response · Authors · 2024-11-29
**Paper update summary**

To all the reviewers,

We have just updated the paper, we thank the reviewers for providing us valuable feedbacks. We report all the changes in this general message, but each reviewer will also find a response specific to their comments in the corresponding section.

The update includes a new experiment on RoBERTa, better experimental setups for ResNets, and many appendices.

All the changes are detailed below:
## Experiments:
The main changes of the paper are the experiments on sequential models:
- **ResNets** [fixed]: as some reviewers mentioned, the ResNet models were far from reaching the same accuracy as the original paper. We have resolved this issue, and now the models have the same performance, architecture and number of parameters as the original paper. There are now two ResNet experiments: a ResNet32 and a ResNet110, both on CIFAR-10. These results have been added to the updated paper. We are considering upgrading one to ImageNet, but due to time constraints this can only be done after the rebuttal.
- **GPT-2** [improved]: We have fixed an error in our code for pre-training the transformer on Wikitext103, which led to improved results. We have also doubled the number of total training steps.
- **RoBERTa** [new]: We introduce a new language modeling experiment: fine-tuning a pre-trained model. We train RoBERTa (124M parameters) on the MNLI task (part of the GLUE benchmark).

Additionally, we have added more figures for the RNN experiments to make the speedups with Highway-BP more clear:
- **Table 3**: a summarized table of the speedups for all RNN experiments.
- **Figure 5**: a plot of loss vs. runtime for the largest RNN model.

## Discussion:
- Many new appendices have been added:
  - **Appendix C**: a very detailed description of experimental setup (models, datasets, hyperparameters, choices of $g_i$ and $r_i$ for each model).
  - **Appendix D**: an analysis of the memory overhead of Highway-BP.
  - **Appendix E**: a discussion about practical considerations for using Highway-BP in a distributed setting, and how it can coexist with existing techniques.
  - **Appendix G**: explanations of how we handle downsampling in ResNets.
- We have reformulated some explanations to make them clearer, and overall improved the readability of the paper.

---

### Meta-Review · Area_Chair_XGtu · 2024-12-14

**Metareview:**

# summary
This paper presents an approach for speeding up training of sequential DNN models by parallelization of the gradient calculation. It proposes to approximate gradient by iterative computation from different residual paths in parallel while pruning some paths. Extensive experiments validate the effectiveness of the proposed method.

# strengths
+ The paper addresses an important problem in deep learning -- how to accelerate training.
+ The idea of iterative gradient propagation along the residual path is novel and interesting.
+ The empirical results are promising.
+ The paper is well written.

# weaknesses
- It has limited benefits: accelerate training at the cost of performance at inference time.
- scalability: it conducted experiments only on small models not LLM
- It is not clear how to generalize to other models.

# decision
Since the paper addresses an important problem in deep learning, and the proposed method is novel and interesting with promising experimental results, I recommend accept.

**Additional Comments On Reviewer Discussion:**

Overall, this paper receives positive reviews. The reviewers find the technical novelty and contributions are significant enough for acceptance at ICLR. Although the reviewers had some concerns on the experimental settings and how to scale and generalize to other models, the authors's rebuttal and updated manuscript help addressed the above concerns.

---

### Decision · Program_Chairs · 2025-01-22

Accept (Oral)